# Targetable lesions and proteomes predict therapy sensitivity through disease evolution in pediatric acute lymphoblastic leukemia

Amanda C. Lorentzian[1,2], Jenna Rever [1,2], Enes K. Ergin [2,3], Meiyun Guo[1,2], Neha M. Akella[1,2], Nina Rolf [1,2], C. James Lim [1,2], Gregor S. D. Reid [1,2], Christopher A. Maxwell [1,2] ✉ & Philipp F. Lange [2,3] ✉

Childhood acute lymphoblastic leukemia (ALL) genomes show that relapses often arise from subclonal outgrowths. However, the impact of clonal evolution on the actionable proteome and response to targeted therapy is not known. Here, we present a comprehensive retrospective analysis of paired ALL diagnosis and relapsed specimen. Targeted next generation sequencing and proteome analysis indicate persistence of actionable genome variants and stable proteomes through disease progression. Paired viably-frozen biopsies show high correlation of drug response to variant-targeted therapies but in vitro selectivity is low. Proteome analysis prioritizes PARP1 as a pan-ALL target candidate needed for survival following cellular stress; diagnostic and relapsed ALL samples demonstrate robust sensitivity to treatment with two PARP1/2 inhibitors. Together, these findings support initiating prospective precision oncology approaches at ALL diagnosis and emphasize the need to incorporate proteome analysis to prospectively determine tumor sensitivities, which are likely to be retained at disease relapse.

Relapsed cancer is a leading disease-related cause of death for children and adolescents[1]. Targeting the specific molecular changes that arise in cancer cells may improve patient survival[2]; for this reason, clinical trials centered on next generation sequencing (NGS)-based identification of biomarkers and targetable pathways are currently establishing patient enrolment strategies, clinical protocols, and critical safety data for personalized therapies. Most trials are currently limited to high-risk or recurrent disease. However, rapidly progressing disease often limits successful treatment options. In certain cases, precision oncology trials may be initiated at diagnosis. But, a major challenge for prospective precision oncology approaches is our limited understanding of the persistence or evolution of targetable lesions and their associated proteins or pathways, and responses to targeted agents, that may be gained or lost at relapse.

There is now a wealth of publicly available data for genomic and transcriptomic characterization of paired diagnosis and relapse specimens from children with cancer. Clonal evolution does occur in pediatric leukemia wherein a minor clone, present only at low frequencies at diagnosis, is selected at time of relapse[3–5]. While these studies show evolution in single nucleotide variants, often through chemotherapy-induced mutation such as *NT5C2* or *CREBBP*[6], structural variants change less frequently with leukemia progression.

Few studies have yet investigated how protein levels change from diagnosis to relapse, especially pertaining to therapeutic targets. A previous proteomic analysis of matched diagnostic and relapsed B-cell precursor acute lymphoblastic leukemia (ALL) specimens, sourced from pediatric and adult patients, observed increased protein levels in specific pathways at relapse, including glycoloysis, phosphate pentose pathway and metabolic pathways that may contribute to chemo-

¹Department of Pediatrics, University of British Columbia, Vancouver, Canada. ²Michael Cuccione Childhood Cancer Research Program at the BC Children's Hospital Research Institute, Vancouver, Canada. ³Department of Pathology and Laboratory Medicine, University of British Columbia, Vancouver, Canada. ✉ e-mail: cmaxwell@bcchr.ca; philipp.lange@ubc.ca

resistance, but it was not specific to pediatric ALL and was limited to ~1400 proteins[7]. Since proteins are the actual therapeutic targets[8,9], it is crucial to better understand the pediatric tumor proteome to determine how the response to therapy may change through progression.

Here, we present a comprehensive interrogation of the dynamics of ALL proteomes and genomes from diagnosis to relapse in paired patient specimens, specifically to understand how cancer-driving and potentially targetable lesions persist or differentiate through disease progression.

## Results

### Next-generation sequencing (NGS) reveals stability of affected genes through ALL disease progression

To examine genomic evolution in relapsed pediatric ALL, we sourced 25 paired initial diagnosis (Dx) and relapse (R) bone marrow biopsies from 11 pediatric patients seen at BC Children's Hospital (BCCH) (Supplementary Data S1) and publicly available whole-exome sequencing (WES) data from 138 specimens (69 paired biopsies) collected and analyzed by the St. Jude's Children's Research Hospital (SJH) (Fig. 1a, Supplementary Fig. 1).

At time of relapse of pediatric ALL, clonal evolution is frequently convergent, and includes the outgrowth of clones defined by a different mutational site within the same affected gene; for example, a relapse clone with *KRAS*.A146T replaces the diagnostic *KRAS*.G12D clone[3]. Since alternate pathogenic mutations in the same gene often serve as biomarkers for the same targeted treatment, we focused our analysis on the affected gene, rather than the mutation site. We first explored the mutational landscape in paired progression samples (*n* = 10 B-ALL, *n* = 1 T-ALL) in the BCCH cohort via targeted, pediatric cancer-focused NGS analysis and reported all somatic mutations identified[10]. We detected recurrent copy number variants (CNV) or single nucleotide variants (SNV) in *CDKN2A/B, NRAS, KRAS, IKZF, JAK1,* and *JAK2* (Supplementary Fig. S2, Supplementary Fig. S2, and Supplementary Data S2), which are commonly mutated in pediatric ALL samples[9]. Nine of the eleven patients had at least 50% retention of mutations and four of these patients had 100% retention of mutations (Supplementary Fig. S2).

We evaluated the lesions grouped by detection only at diagnosis (Dx unique), only at relapse (R unique), or at both timepoints (shared). Here we found 67% (30 of 45) of affected genes were shared between paired diagnosis and relapse samples in the BCCH cohort (Fig. 1b). To determine the generalizability of this finding, we mined all mutational findings from an additional cohort: public NGS data from ALL cases (*n* = 49 B-ALL; *n* = 20 T-ALL) treated at St. Jude's Hospital (SJH) (Supplementary Data S3, Supplementary Fig. S4)[11]. Samples collected from either the BCCH cohort or the SJH cohort showed similar distributions of affected genes that were shared between time-points or were unique to Dx or R (Fig. 1b and Supplementary Data S4), with the majority of variants shared between paired diagnosis and relapse samples (Fig. 1c). In fact, the genes that were persistently mutated through disease progression were highly similar in both cohorts, including *CDKN2A/B, IKZF1,* and *N/KRAS*, with structural variants being retained with higher frequency than SNVs (Supplementary Fig. S5a–c). *NT5C2* mutations were detected only at relapse in the SJH cohort (10% of relapse samples), but were not detected in the BCCH cohort. Surprisingly, retention of genetic lesions was not correlated with the time between diagnosis and first relapse, or between relapses (Fig. 1d and Supplementary Fig. S5d, e). However, we observed relatively higher lesion stability for Hyperdiploid (hyper), Philadelphia + (PH), and early T-cell precursor (ETP) sub-types (Fig. 1e).

### Matched patient-derived leukemic cells respond similarly to variant-selected agents

Persistence of affected genes suggests that sensitivity to precision therapies may also persist with disease progression. To examine this,

we paired affected genes with targeted agents following the Pediatric MATCH strategy and evidence from clinical trials or case reports, as described[10]. For the combined analysis of the SJH and BCCH cohorts (*n* = 80 paired samples), we found 64% of ALL patients (51 of 80) retained at least one variant-agent pairing at disease relapse; in fact, nearly 50% of patients (38 of 80) showed complete retention of variant-agent pairings through disease progression (Fig. 1f and Supplementary Data S5). The target with the highest retention was *CDKN2A* deletions paired with CDK4/6 inhibitors and shared in 68.9% of occurrences. *NRAS* and *KRAS* mutations paired with MEK inhibitors were also highly retained and were shared in 57.1% and 61.5% of occurrences respectively, although *NRAS* mutations occurred in 21 patients compared to 13 for *KRAS* (Fig. 1f). Conversely, for the five patients that had MTOR targets (*PIK3CA, PTEN,* or *MTOR* mutations), none of the targets were shared. Only five patients had variant-agent pairings unique to the diagnostic timepoint, while 13 patients harbored no targetable mutations (Fig. 1f).

We sourced viably-frozen bone marrow mononuclear cells (BM-MNC) from paired progression events for six patients treated at BCCH, including four paired BM-MNC that showed retention of all affected genes and two paired specimens with partial/no retention. Within this cohort, we identified four targeted agents (*CDK4/6* pathway mutations: Palbociclib; *MEK* pathway mutations: Trametinib, *JAK/STAT* pathway mutations: Ruxolitinib; *SMO* pathway mutations: Vismodegib) and then we treated viable patient specimens with graded doses of these agents. We used image-based drug screening of B-ALL cells co-cultured with mesenchymal stromal cells (MSC) to determine IC50 values for each sample after exposure for 48–72 h to the four different targeted agents (Fig. 1g). Overall, the measured IC50 values revealed poor selectivity, with predicted responses to trametinib being an exception. Trametinib IC50 values were lower for patient samples predicted to be sensitive to MEK inhibition (Fig. 1g), although the difference was not significant (*p* = 0.13). In contrast, Palbociclib IC50 values did not differ within the primary ALL cohort (Fig. 1g), suggesting cytotoxicity is induced in a non-targeted manner by this agent. Overall, the measured IC50 values in relapsed samples correlated with values measured in matched diagnostic samples (Pearson's *r* = 0.85, *p* = 6.0e−7, Fig. 1h). Therefore, our genomic analysis of pediatric ALL disease progression samples revealed stability of gene lesions that are known therapeutic targets. Drug sensitivities within matched diagnostic and relapsed samples were also highly correlated, though the drugs showed poor selectivity.

### Global proteome analysis shows stability through progression and groups cases with poor outcome

To determine whether the observed persistence of cancer-associated targetable genomic lesions and associated drug sensitivities is also characteristic of the proteome, we next conducted a comprehensive analysis of 48 primary (*n* = 39 B-ALL, *n* = 9 T-ALL) specimens from Dx and R sourced from the BCCH biobank. This proteomics cohort included 14 specimens from six patients with matched biopsies taken at diagnostic and subsequent relapse timepoints that were also in the BCCH genomics cohort (Supplementary Fig. 1). In addition, we included five ALL cell lines in our study (B-ALL = 4, T-ALL = 1). Our diverse cohort span the major cytogenetic groups and ages ranging from 2 years to 23 years (Fig. 2a). Male patients were moderately over-represented (65% male compared to 35% female) (Fig. 2a, Supplementary Data S1).

We employed a data independent acquisition approach (DIA) using a study specific spectral library of 10,130 proteins to quantify 8,590 proteins (Supplementary Fig. S6, Supplementary Data S6 and S7). To determine if the proteome distinguishes leukemia types, non-cancer monocytes and cell lines, we first filtered for proteins with highly variable protein abundance across samples (Supplementary Data S8). Highly variable proteins were selected by calculating the

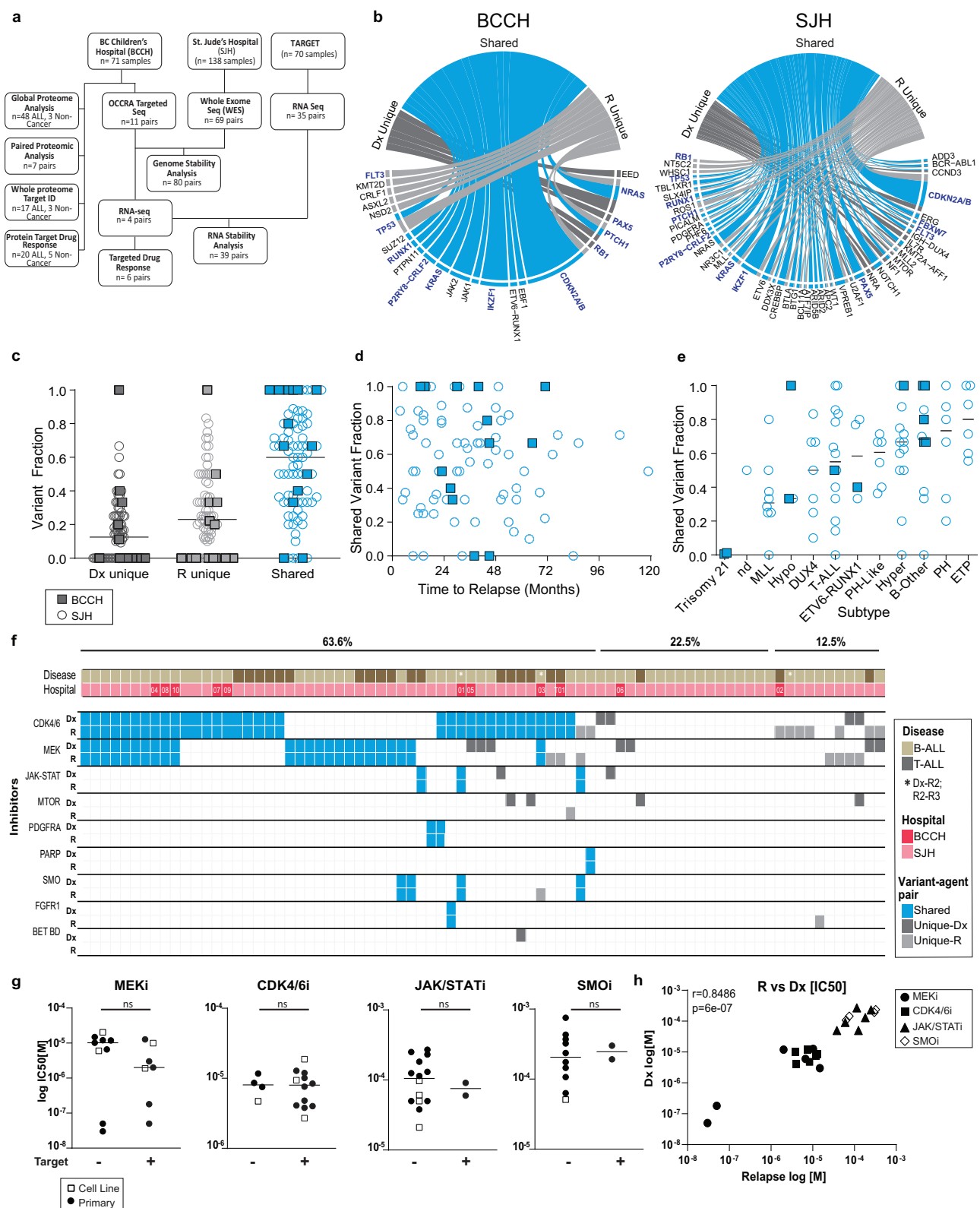

log2 fold-change for each protein relative to the median protein abundance across all samples for that protein. Only the proteins with a log2FC > 2 in at least five samples were selected for further analysis (3907 proteins). Samples clustered distinctly by B-ALL, T-ALL and non-cancer monocytes and cell lines (Fig. 2b). Principal component analysis (PCA) demonstrated that age and sex did not contribute significantly to the variation in the data and the variation is most driven by sample type (Supplementary Fig. S7). In line with earlier

reports, B-ALL cell lines cluster away from primary samples[12,13] suggesting phenotypic differences, thus highlighting the importance of direct studies of primary samples. Interestingly, there was one cluster comprised of several different sample types. Upon further investigation of blast percentage, we found this cluster to consist entirely of low-blast samples (the three non-cancer BM specimens, two low-blast T-ALL samples and one low-blast B-ALL sample). These findings demonstrate the sensitivity of our proteomics analysis to identify

**Fig. 1 | Stability of affected genes and targeted drug response through ALL disease progression. a** Flow-chart depicting the number of samples from each cohort that were included for each analysis. **b** Circos plot for all mutations identified in the BCCH and SJH cohorts demonstrating mutations that were Dx unique (dark gray), R unique (light gray), or shared between Dx and R samples (blue). Genes in blue text indicate genes with detected lesions in both cohorts. **c** Fraction of variants identified as D unique, R1 unique, or shared within paired samples sourced from 80 ALL patients ($n = 11$ patients from BCCH represented by squares, $n = 69$ patients from SJH represented by circles). The black bar represents the median of the population. **d** Dot plot for fraction of shared variants versus time to relapse for 80 ALL patients ($n = 11$ patients from BCCH represented by squares, $n = 69$ patients from SJH represented by circles). **e** Dot plot for fraction of shared variants separated by disease subtype for 80 ALL patients ($n = 11$ patients from BCCH represented by squares, $n = 69$ patients from SJH represented by circles). The

black bar represents the median of the population. **f** Predicted sensitivity to targeted agents in paired Dx-R samples (or Dx-R2, R2-R3 indicated by an asterisk) taken from 80 ALL patients treated at BCCH ($n = 11$ patients, red) or SJH ($n = 69$ patients, dark pink). B-ALL (light brown) and T-ALL (dark brown) samples are indicated. Shared variants (blue), Dx unique variants (dark gray), or R unique variants (light gray) are indicated. Agent-variant pairs were assigned following the strategy outlined in the Pediatric Match Trial[9]. **g** Dot plots for IC50 [μM] values measured for primary samples or cell lines for each of four inhibitors. Measurements represent the mean of $n = 2$ replica wells from a single experiment. Samples are separated based on the presence or absence of a genomic variant predicted to augment drug sensitivity (ns= not significant by unpaired, two-sided $t$-test). **h** Correlation of IC50 [M] values measured for paired Dx and R samples. Individual drugs are indicated by unique identifiers. r = Pearson correlation coefficient.

---

biological differences between sample types and perform unsupervised classification.

To better characterize our largest patient group, we conducted a focused analysis of the samples in the two B-ALL clusters. Unsupervised hierarchical clustering of proteins with high variability in B-ALL (Supplementary Data S8) resulted in seven proteome clusters (Fig. 2c). Paired samples cluster closely together for four of six patients that had multiple timepoints (Fig. 2c) indicating high similarity consistent with our genomic findings. As well, some cytogenetic subtypes showed stronger trends in co-clustering, for example, cluster P3 primarily contained ETV6-RUNX1 patients. Hypodiploid patient samples also clustered closely together although spread across two clusters. Cluster P4 was the largest cluster and consisted almost entirely of BCP-ALL or "other" samples, indicating similar proteomes although these samples are not characterized by a major shared genome alteration. The remaining clusters were a mixture of subtypes, suggesting phenotypic similarities across cytogenetic subtypes (Fig. 2c).

Gene ontology enrichment analysis identified distinct biological processes with differential protein abundance between the clusters (Fig. 2c, Supplementary Data S9). Notably clusters P5, P6 and P7 have higher abundance of proteins involved in antigen presentation and leukocyte activation (cluster P3). Cluster P5 was enriched for processes related to actin and cytoskeleton organization while cluster P2 had the highest abundance in proteins relating to humoral immune response.

High-risk cases were associated with all clusters but enriched in cluster P4. Interestingly, stratification by risk group did not yield significant differences in 5-year event-free survival (event = relapse or death) (Fig. 2d). However, stratification by unsupervised proteome-cluster followed by Kaplan Meier analysis of the major proteome-clusters showed significant differences between clusters P2, P3, P7 and cluster P4, which was associated with a high event rate (Fig. 2e). Overall, our findings indicate phenotypic differences that are not solely linked to the common ALL cytogenetic subtypes, and highly similar proteomes between paired patients, consistent with our observation of genomic stability.

### Cancer-associated proteins and processes remain stable through disease progression

To interrogate the apparent similarity observed in patient-matched progression samples, we further examined the proteomes for the 6 pairs that had matched Dx-R or R-R biopsies (BALL01, BALL03, BALL04, BALL05, BALL06, BALL07) plus one additional PDX-expanded Dx-R-R set (BALL02) (Supplementary Figs. S8 and S9, Supplementary Data S10 and S11). To better understand inter- and intra- patient stability among the disease states, we tested proteins for statistically significant equivalence in all possible patient and timepoint pairings by two-one-sided t-test (TOST) and corrected for a 5% false discovery rate (FDR). As expected, the rate of significantly equivalent proteins was lowest when proteomes of non-cancer specimens were compared to proteomes of cancer specimens (63% or 68% equivalent to Dx or R,

respectively) (Fig. 3a, Supplementary Fig. S10a). In contrast, >90% of robustly quantified proteins showed equivalent abundance when comparing proteomes of matched diagnosis and relapse specimens, or multiple relapses, from the same patient (Fig. 3a, Supplementary Fig. S10B, Supplementary Data S12). Only BALL05 and BALL06 showed low equivalence (59% and 69% respectively) (Fig. 3a), and these paired samples showed similar variation in gene mutations (Fig. 1f) and proteome cluster analysis (Fig. 2c). At only 75% (median equivalence), diagnosis or relapse samples obtained from different patients show significantly lower proteome equivalence than matched samples from individual patients through progression (Fig. 3a).

To determine processes that are particularly stable throughout progression, we next performed a gene set and pathway enrichment analysis. Proteins found to be equivalent between cancer and non-cancer were removed prior to the enrichment analysis to eliminate 'housekeeping' mechanisms that are generally stable. Pathway enrichment analysis identified processes linked to overall cell survival as equivalent amongst cancer proteomes, including transcription related processes, metabolic processes, and cellular responses to DNA (Fig. 3b and Supplementary Data S13). Investigation of proteins that were statistically different between the pairs did not reveal any processes as significantly enriched between diagnosis and relapse. Instead, we found protein abundance differences between time points to be patient-specific (Supplementary Fig. S11, Supplementary Data S14). We next probed proteins involved in B-cell development, such as transcription factors IKZF1, EBF1, PAX5, VPREB1, and TCF[14] (Supplementary Data S15), that are commonly dysregulated in ALL. Overall, abundance of these proteins was significantly higher in cancer samples than in mature B-cells isolated from non-cancer peripheral blood mononuclear cells (PBMCs) (Fig. 3c, Supplementary Fig. S12). Moreover, for most patients, stable abundance was observed between disease states (Fig. 3c, d).

Finally, we identified 45 cancer-associated proteins (CAPs) to be significantly more abundant in diagnosis specimens ($n = 6$) or relapse specimens ($n = 12$) compared to non-cancer controls ($n = 3$) (Supplementary Fig. S13a-d and Supplementary Data S16), including several proteins that are commonly overexpressed in acute lymphoblastic leukemias such as FLT3, CDK6, and EBF1[15]. Given that the 45 CAPs are significantly more abundant in the ALL specimens than in the non-cancer specimens, they are likely linked to tumorigenic processes in our samples and would be of interest to determine their stability through disease progression. The majority ($n = 36$ proteins) showed increased abundance at both disease time-points (Fig. 3e) and their abundance levels were positively correlated between diagnosis and relapse (or relapse-relapse) (Pearson's $r = 0.75$, $p < 2.2e-16$); similarly, proteins with lower abundance ($n = 10$ proteins in either disease state) showed significant positive correlation between disease states (Pearson's $r = 0.70$, $p < 3.5e-15$) (Supplementary Data S17 and Supplementary Fig. S13e−f). Further investigation of the few outliers revealed that FLT3 was commonly over-abundant at Dx and lower abundant at

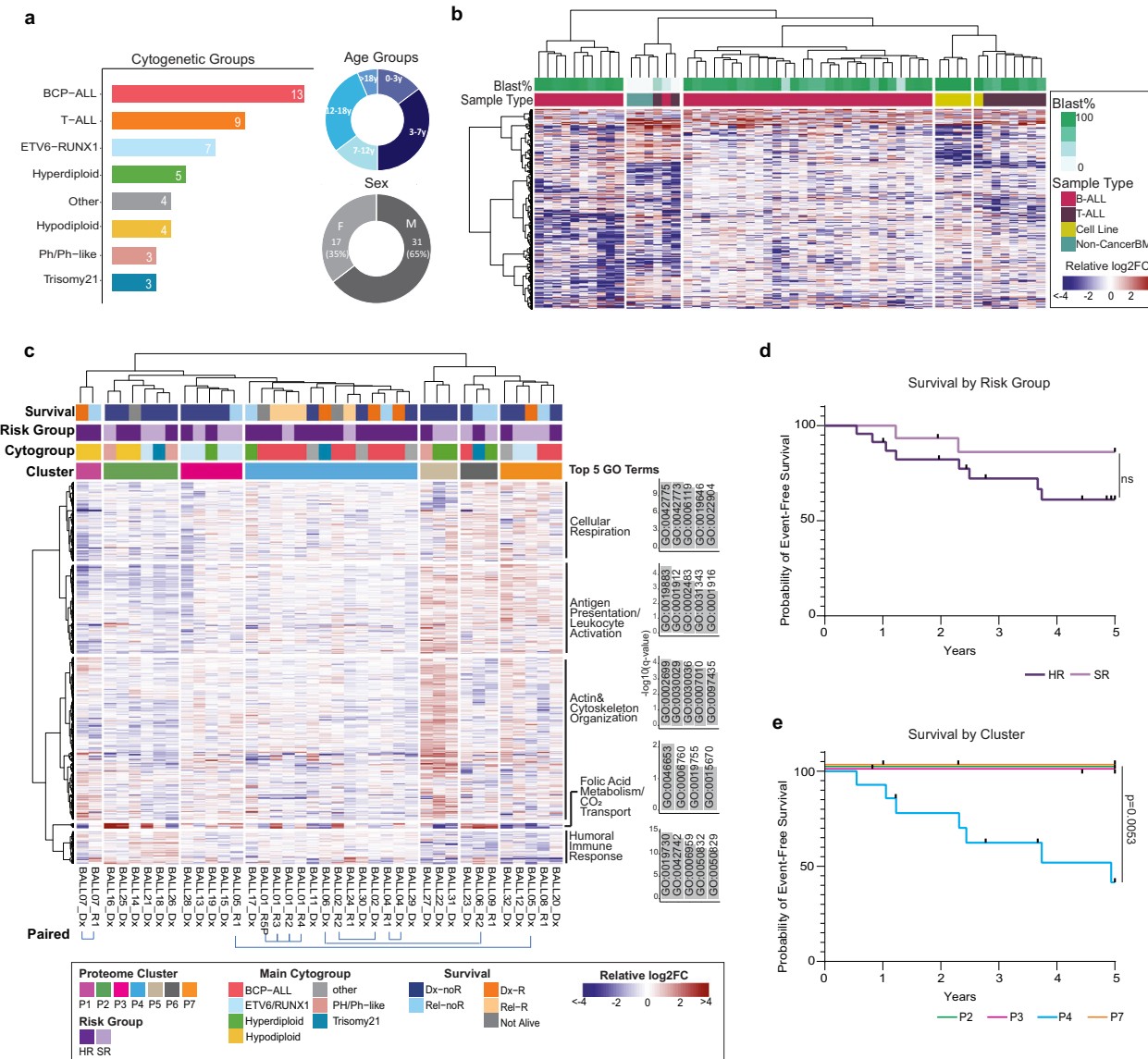

**Fig. 2 | Global proteome analysis shows stability through progression and groups cases with poor outcome. a** Descriptive summary of the cohort for proteome analysis. The bar plot represents the total number of each cytogenetic subtype and the donut plots represent the age (top) and sex (bottom) of the patients. **b** Hierarchical clustering of 3907 variable proteins represented by the relative log2FC (protein intensity/median protein intensity). The color bars indicate sample type (bottom) and leukemic blast percentage (top). **c** The major B-ALL clusters were selected for in-depth characterization. Hierarchical clustering of 935 proteins based on log2FC defined seven sample clusters (horizontal) and five protein clusters (vertical). The other color bars indicate main cytogenetic subgroup (second from the bottom), followed by clinically assigned risk group (SR = standard risk, HR = high risk), and current survival status on top. The five most

significant GO terms for each cluster of proteins were selected for visualizations. The annotation to the right of each cluster of proteins is the summary of the top significant terms. Bars represent the adjusted p-value for each GO term. **d** Kaplan Meier survival curve with up to 5 year follow-up data for B-ALL clusters (n = 38) grouped by clinically assigned risk group (SR = standard risk, HR = high risk), ns = not significant by logrank test for trend. Black tick marks on the survival curve represent data that has been censored due to follow-up data <5 years. **e** Kaplan Meier survival curve with up to 5 year follow-up data for B-ALL samples grouped by proteome cluster for clusters with >4 samples (n = 30). Significance assigned by logrank test for trend. Black tick marks on the survival curve represent data that has been censored due to follow-up data <5 years.

Relapse (observed in BALL 01R4-R5P, BALL02, BALL03, BALL05) suggesting the loss of FLT3 may be a relapse-specific mechanism. Other outliers were patient-specific, such as the higher abundance of SELP at Dx for BALL04. Restricting the comparisons to matched specimens (n = 7 patients) confirmed that the high correlation was retained at the level of individual patients (Fig. 3f, Pearson's r = 0.67 − 0.90), indicating the stability observed from disease progression across the global proteome is also observed when restricting the analysis to significantly more abundant CAPs. To determine if this high level of stability is similar at a transcript level, we performed amplicon-based transcriptome sequencing for four paired Dx-R patients (BALL01,

BALL04, BALL06, & BALL07) (Supplementary Data S18). BALL06 had the lowest correlation, consistent with low gene mutation and protein level stability observed in these samples. But, the other three progression pairs were highly correlated (Pearson's r > 0.85) and clustered together by hierarchical clustering (Supplementary Fig. S14a). This conclusion was confirmed with the analysis of a larger publicly available dataset (TARGET)[16], which included paired Dx and R samples from 35 patients with pediatric ALL (n = 70 samples) (Supplementary Data S19). Hierarchical clustering of samples based on Pearson correlation revealed 51% of pairs clustered either as neighbors (34%) or within the same cluster (17%) (Supplementary Fig. 14b). Similar to our proteome

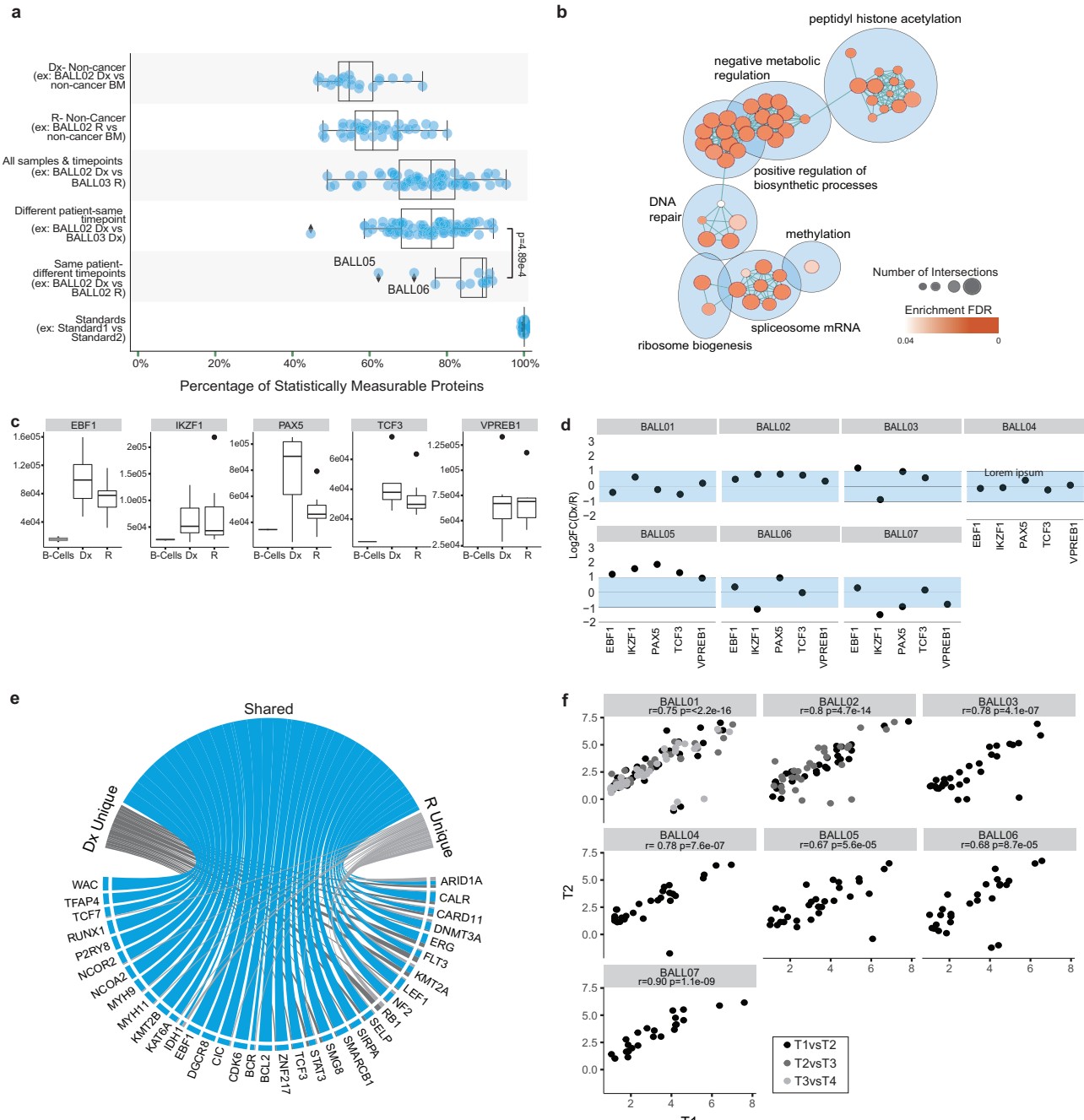

**Fig. 3 | Cancer-associated proteins and processes remain stable through disease progression. a** Summary of tests for equivalence (Two-one-sided t-test (TOST) for equivalence, boundaries between log2FC < −1 and log2FC > 1, FDR < 5%) of protein abundance between different groups and pairings. Only statistically measurable proteins are represented. Each dot represents the mean equivalence or difference of all protein abundance for a pairing. The Mann–Whitney *U* test (two-sided) was used to test the difference between the two groups' percent equivalence boxplots (same patient-different timepoint vs different patient-same timepoint). The number of comparisons in each group from top to bottom are *n* = 21 pairs, *n* = 45 pairs, *n* = 76 pairs, *n* = 98 pairs, *n* = 17 pairs, and *n* = 210 pairs. Box represents the interquartile range (IQR), the middle line represents the median and the whiskers extend to 1.5 × IQR. **b** Pathway enrichment analysis of the stable population of proteins. The color of the circles indicates the enrichment FDR and size represents the number of identifications for the term. **c** Abundance of transcription factors of interest for each sample separated by timepoint (T1 or T2, black) compared to protein abundance in mature B-cells (*n* = 2 samples) isolated from peripheral blood mononuclear cells (gray). Box represents the IQR, the middle line

represents the median and the whiskers extend to 1.5 × IQR. **d** Dot plots represent the log2FC of timepoint 1(T1)/timepoint 2(T2) for each of the proteins for each sample. The shaded blue area indicates the stable range of −1 to 1FC. *For patients with multiple timepoints (BALL01 and BALL02) only the log2ratio of the earliest timepoint/the latest timepoint is represented for simplicity. **e** From the list of 269 pediatric cancer-associated proteins (CAPs), 141 proteins were detected in our data and 45 proteins were deemed significant (LIMMA analysis of Dx samples vs. non-cancer bone marrow (BM) samples and R vs non-cancer BM samples (log2FC > 1, *p*-value adjusted BH-FDR < 0.05). Circos plot summarizes significantly over-abundant cancer-associated proteins (CAPs). **f** The protein abundance for each protein that was over-abundant at Dx (*n* = 22 proteins), was plotted as timepoint 1 (T1) vs timepoint 2 (T2), where T1 is the earliest timepoint available (log2(protein expression/the average protein expression in the non-cancer BM)). Pearson's *r* correlation was calculated for all sample pairs. In cases of multiple time-points the correlation was calculated for consecutive pairings and are represented by the different colored dots.

findings of stability in cancer-associated proteins, correlations for expression of cancer-associated gene products (mRNA) for each patient (BCCH and TARGET) were also highly stable with 82% of pairs having a Pearson's $r > 0.85$ (Supplementary Fig. 14c).

### Whole proteome discovery-driven analysis identifies pan-ALL protein targets

The measured cytotoxicity was disappointing for agents informed by the Pediatric Match genetic variant-agent prioritization strategy (Fig. 2c) leading us to probe our proteome datasets for alternative targets. To discover pan-ALL protein targets, we first filtered for proteins identified in more than 40% of specimens and with a high overall abundance (log10 intensity) and strong abundance increase over non-cancer (log2 FC) (cut-off: at least 95th percentile for both metrics) (Fig. 4a). We defined stable protein abundance between paired diagnostic and relapsed samples as a model variable for target discovery; using this criterium, and representing patient BALL01 as an example (Fig. 4b), we generated a ranked list of pan-ALL targets, which included HSPB1, PARP1, and PRDX1 as top-ranked candidates (Fig. 4c). We selected to further characterize PARP1 as a candidate target since PARP1/2 inhibitors are already developmental therapeutics for a variety of pediatric tumors[17]. The efficacy of PARP1/2 inhibitors has also previously been demonstrated in AML[18], ALL cell lines[19], and other preclinical leukemic models[20], however their usage for pediatric ALL with elevated PARP abundance has not yet been shown. In addition, PARP1 is activated by DNA damage as a repair mechanism[21], and "cellular responses to DNA damage" showed enrichment in our prior pathway enrichment analysis, providing further confirmation that this pathway is overexpressed and stable (Fig. 3e).

To validate the hypothesis that PARP1 elevation is reflective of an increased dependency on DNA repair in response to DNA damage, we examined the cellular response to genotoxic stress in viably-frozen ALL cells ($n = 3$ patients) or non-cancer bone marrow-derived stem cells (BMSC) ($n = 2$ donors). We co-cultured primary cells on hTERT-MSCs for 24 h without (sham) or following exposure to genotoxic ionizing radiation (1 Gy X-ray) (Fig. 4d). After 30 min or 24 h to allow induction or resolution of damage respectively, expression of the DNA damage marker gamma-H2AX or PARP1 was examined by immunofluorescence and the intensity of staining was normalized to the baseline levels measured at 30 min in sham treatment (Supplementary Fig. S15a). This analysis revealed an expected increase of gamma-H2AX foci (DNA damage) 30 min after X-radiation in both BMSC and B-ALL samples (Fig. 4e, Supplementary Fig. S15B). The number of gamma-H2Ax foci was significantly elevated at 30 min after 1 Gy X-radiation in B-ALL samples relative to control BMSCs (Fig. 4e) potentially indicating their hypersensitivity to genotoxic stress. The number of gamma-H2AX foci was reduced in both populations by 24 h following X-radiation (Fig. 4e), albeit to a lesser extent in B-ALL cells, indicating either repair or clearance of damaged cells. However, the pattern of PARP1 expression was distinct for B-ALL cells relative to BMSC, increasing significantly in response to X-radiation (measured at 30 min) as well as following proliferative stress (at 24 h in sham) (Fig. 4f)(Supplementary Fig. S15c), suggesting a reliance on PARP1 expression for B-ALL cell survival following stress.

To test a possible reliance on PARP1 for survival and its potential suitability as a target for therapeutic intervention we treated primary patient specimens with graded doses of two PARP1/2 inhibitors, Olaparib and PJ34. We sourced viably-frozen ALL samples ($n = 18$) from the BCCH Biobank, including matched specimens used in our discovery cohort ($n = 4$), additional ALL specimens ($n = 13$), and non-cancer pediatric stem cell samples ($n = 5$). Image-based drug screening of ALL cells co-cultured with MSC demonstrated high cytotoxic specificity of PARP1/2 inhibitors for ALL relative to non-cancer BMSC samples, as determined by IC50 values for each sample (Fig. 4g, h). To investigate whether this result is an effect of increased cell proliferation, we

measured the percentage of phospho-histone H3 positive cells in culture. We found, however, <2.5% of cells in the in vitro drug screening assays are mitotic (Supplementary Fig. S16a–b) suggesting cell proliferation is not the target for PARP inhibition. We next measured the protein abundance of key mitotic and cell cycle regulators shown to differentiate cells in G2 and M phases[22]. While the expression level of PARP1 was universally higher in B-ALL samples, we found the levels of key mitotic and cell cycle regulators were not elevated in Dx and R samples relative to non-cancer BM controls (Supplementary Fig. S16c, d). The significant increase in sensitivity of ALL cells to PARP1/2 inhibitors relative to non-cancer cells indicates that this may be a potential pan-ALL therapeutic target that was discovered through protein abundance analysis.

## Discussion

Molecularly guided targeted therapies have a high potential to improve outcomes for pediatric cancer patients. Yet, only 3–58% of patients receive molecularly guided therapies and even fewer report a positive response to treatment in NGS-guided trials[23]. These unsatisfactory outcomes can be ascribed to multiple limitations, including the reliance on genomics for target identification, which cannot capture the plasticity of downstream transcriptional, translational and post-translational processes that impact target abundance and drug sensitivity, and the restricted enrollment for high-risk or relapsed cancers, which often progress quickly. Several initiatives, including ZERO and INFORM, have recently exemplified the use of precision oncology at diagnosis to help refine, or even change subtype diagnosis for several cancer types, which can lead to more appropriate treatment options[24,25]. It is actively debated whether initiating molecular analyses for precision oncology at diagnosis is beneficial[23]. Here, we advance the debate by contributing additional evidence of high retention of potential drug targets in pediatric ALL. We show that the stability extends to and may even be more pronounced at the protein level and that proteome analysis can inform target selection in addition to and independent of genomic analysis.

One challenge to initiating precision medicine at diagnosis is the prospect that the dominant relapse clones contain distinct mutations and unique drug sensitivities. Genomic analysis of paired B-ALL samples has inferred clonal structure and evolution through disease progression[3,26–28]; thus, for example, a transition from a major clone at diagnosis carrying a KRAS.G12D mutation is distinguishable from a major clone at relapse carrying a KRAS.A146T mutation[3]. Our analysis did not focus on the gain/loss of clone-defining mutations. Rather, we measured the durability of actionable genomic mutations and stable proteomic features over the course of the disease; in the example of a transition for KRAS.G12D to KRAS.A146T highlighted above, for instance, our analysis did not distinguish between site-specific mutations in the same actionable target. In our cohort, mutations in IKZF1, KRAS and NRAS were persistent at 80%, 75% and 60% of samples, although these have been described as relapse-enriched mutations[3,29]. Although indeed, we identified IKZF1 and KRAS mutations that were unique to relapse, we found as well that if the mutation was detected at Dx it persisted to relapse 100% of the time.

We show that in primary ALL that progresses (80 patients), 64% of patients retain at least one potential drug target at relapse, although it is estimated that 37% of primary tumors retain druggable events at relapse[23]. This high level of persistence was further reflected in the overall correlation of drug sensitivities between diagnosis and relapse. In our assay, genome variants predicted sensitivity to a targeted MEK inhibitor, which is consistent with prior studies[26,27]. However, we found no correlation between the presence of CDKN2A deletion in primary ALL samples and sensitivity to CDK4/6 inhibition. Indeed, the utility of CDKN2A deletion to act as a predictive biomarker for sensitivity to CDK4/6 inhibitors is currently unresolved[28,30–32]. Taken together, our data supports the notion that common genomic biomarkers are not

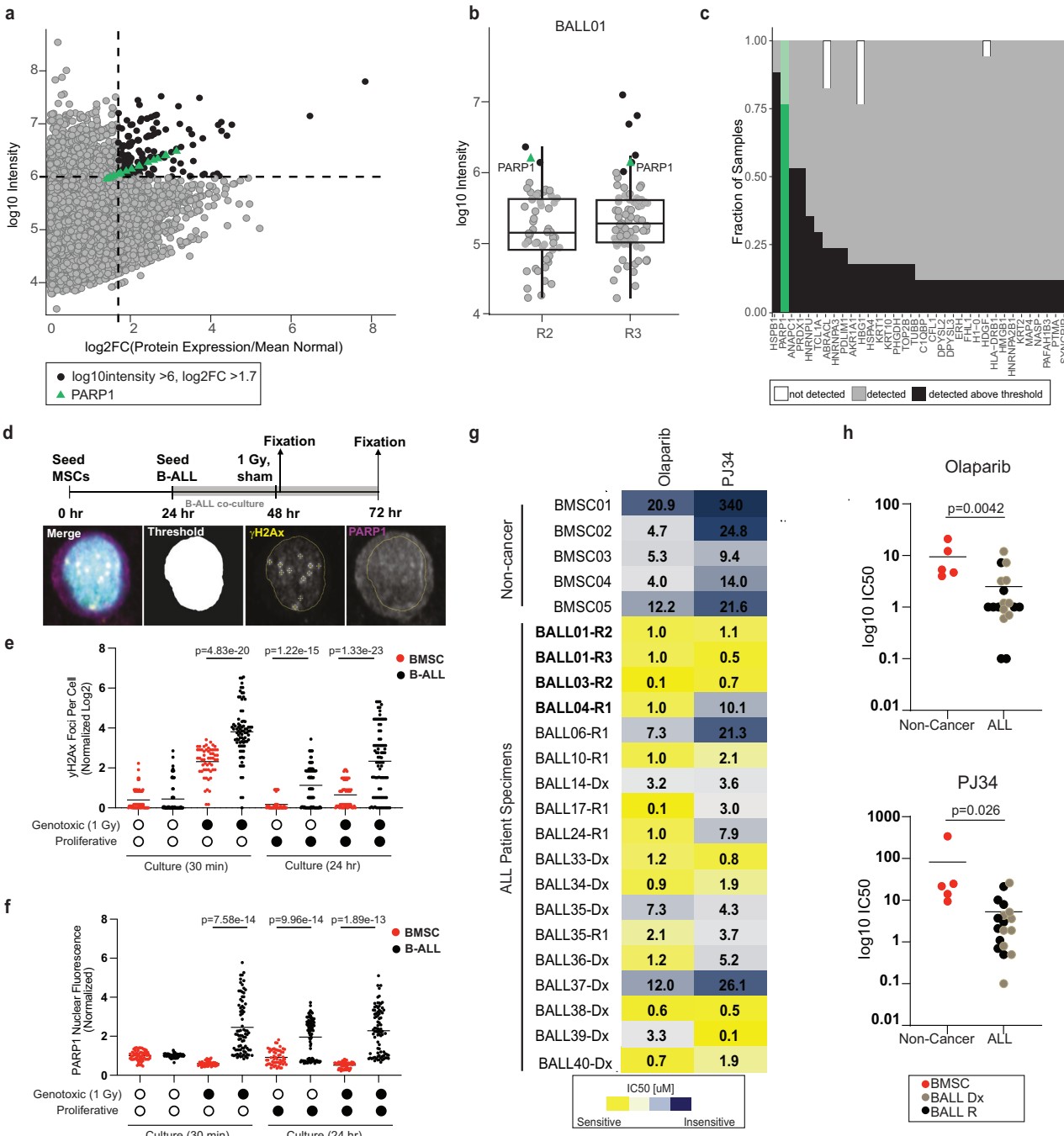

**Fig. 4 | Whole proteome discovery-driven analysis identifies pan-ALL protein targets. a** Correlation of log2 fold-change (FC)/non-cancer vs protein abundance for all proteins in samples from the paired Dx-R dataset. Dashed lines represent cut-offs for top five percent of the population. Proteins that meet both cut-offs are black and PARP expression is represented by green triangles. **b** Representative figure showing all proteins that are >log2FC of 1.7. Only the proteins that have log10 intensity >6.0 (top 5%) are in green. $n = 1$ experiment from the two samples of patient BALL01. Box represents the IQR, the middle line represents the median and the whiskers extend to 1.5 × IQR. **c** All proteins of interest plotted by percentage of samples the protein meets the indicated parameters (black), was identified but not meeting the parameters (gray) or not detected (white). PARP1 is highlighted in green. **d** Experimental timeline and protocol (above) with image analysis pipeline (below) for primary B-ALL and BMSC cocultures followed by immunofluorescence analysis to quantify γH2Ax foci per cell and PARP1 nuclear fluorescence. **e** Log2 γH2Ax foci per cell normalized to sham treatment at 30 min, quantified from immunofluorescence analysis of 2 BMSC (red) samples and 3 B-ALL (black) samples

($n = 30$ cells per sample). Significance is assigned by unpaired, two-sided Welch's $t$-test. **f** Average PARP1 nuclear fluorescence per cell normalized to sham treatment at 30 min, quantified from immunofluorescence analysis of 2 BMSC (red) samples and 3 B-ALL (black) samples ($n = 30$ cells per sample Significance is assigned by unpaired, two-sided Welch's $t$-test. **g** Measured IC50 values for Olaparib or PJ34 measured against ALL or non-cancer samples from patients treated at BCCH ($n = 5$ non-cancer, 8 diagnostic samples, 10 relapse samples). IC50 [μM] are colored by most sensitive in yellow to least sensitive in blue. Measurements represent the mean of $n = 2$ replica wells from a single experiment. Bolded Patient IDs indicate patient samples analyzed in the pan-ALL target proteomic analysis. **h** Dot plots for IC50 [μM] values for Olaparib or PJ34 measured against non-cancer specimens ($n = 5$) (red), primary diagnostic specimens ($n = 9$) (light brown) or primary relapse specimens ($n = 9$) (black). Points represent the mean of $n = 2$ replica wells from a single experiment for each specimen. Significance is assigned by unpaired, two-sided student's t-test.

sufficient to predict tumor sensitivity to variant-targeted monotherapies[33].

To supplement NGS-guided target identification, some precision oncology initiatives are complementing genome analysis with additional molecular strategies such as RNA-seq or Methyl-seq[23,24,34]. Our study conducted comprehensive targeted and non-targeted, paired Dx and Relapse proteome analyses of pediatric ALL. Global quantitative proteome analysis identified clusters that span across established subgrouping based on cytology and risk with one cluster being particularly associated with poor 5-year event free survival. This observation further adds to the emerging notion that proteome-based molecular subtyping has the potential to identify larger groups that are more reflective of the actionable phenotype[35].

Combining tumor proteome insights with genomic data enables an even deeper understanding of disease progression. For example, targeted sequence analysis indicated considerable evolution in BALL03, but the CAP proteome analysis was highly stable ($r = 0.78$, $p < 4.1\mathrm{e}{-07}$). Similarly, the relapse BALL04 sample gained a *TP53* mutation, but CAP stability, proteome stability (equivalence of 90%), and drug responses were highly correlated with the diagnostic sample. Thus, the evolution of minor clones through disease progression may not dramatically impact the expressed proteomes.

Proteomic analysis can also reveal potential targets[36–38]. Consistently, we identified PARP1, PRDX1, ANAPC1 and HSPB1 as overexpressed pan-ALL target candidates. PARP sensitivity has previously been linked to markers of ongoing replicative stress in AML cells[18], and we find similar evidence of ongoing replicative stress in primary B-ALL sensitizing to PARP inhibition. Mechanisms to overcome such stressors enable leukemic cells to maintain active cell proliferation and prolong cell survival[39–41]. Moreover, we validated the sensitivity of B-ALL samples to PARP1/2 inhibition in vitro. Similarly, the other top-ranked, proteome-based, pan-ALL targets warrant further investigation of their actionability in pediatric ALL. The pro-survival antioxidant activity of PRDX1[42] and anti-apoptotic activity of HSPB1[43] may, like PARP1, be necessary for B-ALL cell survival. Overexpression of these proteins appear to be biologically relevant and may be of interest for further exploration.

Our retrospective design was useful for studying matched Dx-R samples, which are relatively rare and difficult to predict prospectively. We note that the conclusions drawn from our proteome analyses and image-based drug screening are limited by the number of matched patient samples examined, the variety of genetic subtypes, and the ethnic diversity of the patients treated in our single site cohort. In addition, image-based drug screening of ex vivo ALL-MSC co-cultures may not accurately reflect responses in patients although it has been shown to capture leukemia-intrinsic differences in cell proliferation and survival and, in the case of Venetoclax, ex vivo responses correlate with strong in vivo antileukemic activity[44]. With these potential caveats in mind, the findings from this study demonstrate clear potential utility for prospective proteogenomic variant identification for the targeted treatment of pediatric relapse ALL to be initiated at first diagnosis.

## Methods

### Patient samples and non-cancer controls
Patient specimens were collected by Biobank staff at BC Children's Hospital. Samples were taken with informed consent from patients and their parents during routine clinical care. Sample collection and experiments were performed as approved by the University of British Columbia Children & Women's Research Ethics (H17-01860), and conformed with standards defined in the WMA Department of Helsinki and the Department of Health and Human Services Belmont Report. Sex was taken into consideration of the study design. Sex was self-reported by the patient or the patient's parents.

Mononuclear cells containing leukemic blasts were isolated by Ficoll-Paque PLUS density centrifugation, and then viably frozen and preserved. Aliquots of patient samples, and patient clinical information were de-identified prior to release for this study. Leukemia samples were immunophenotyped at the clinical hematopathology laboratory using established ALL subtype-specific 10-color flow cytometry panels according to clinical standard operating procedures. Patient bone marrow morphology was assessed by hematopathologists and cytogenetics studies were performed by clinical cytogeneticists. Upon receipt of the specimens, patient mononuclear cells were thawed at 37 °C for 1–2 min, washed 1× in warm RPMI-1640 medium containing 10% fetal bovine serum (FBS, Invitrogen, Waltham, Massachusetts, USA) and washed 2× with PBS and stored as $0.5 \times 10^6$–$1 \times 10^6$ cells per cell pellet.

Bone marrow stem cells (BMSC) from five non-cancer individuals were initially collected following routine procedures for bone marrow stem cell transplantation, and remaining material was stored viably with the BCCH Biobank. In addition, we obtained bone marrow mononuclear cells from one healthy individual (non-cancer BM). Finally, for analysis of mature B-cells, PBMCs from five patients that did not have any hematological malignancies that were in a similar age range were combined.

### Cell lines
The following cell lines were used in the study; Nalm6 (M), Jurkat (M), RS411 (F), BV173 (M), 697 (M). All cell lines were purchased from the American Type Culture Collection (ATCC) and have been sequenced with a targeted NGS sequencing panel to confirm identity. ALL cell lines were cultured in RPMI-1640 media supplemented with 10% fetal bovine serum (FBS) and 2 mM L-Glutamine (Gibco, Grand Island, NY) and maintained at 37 °C in 5% $CO_2$. Cell lines were maintained at concentrations recommended by ATCC and passaged every 2–3 days. Cell lines were not tested for mycoplasm. To collect the cells for MS sample preparation, cells were washed 2× with PBS and stored as $0.5 \times 10^6$–$1 \times 10^6$ cells per cell pellet.

### DNA/RNA extraction and sequencing
DNA and RNA extraction were performed using an Allprep (Qiagen) workflow. Library preparation and targeted sequencing was performed using the Oncomine Childhood Cancer Research Assay (OCCRA) on an Ion Chef and Ion Torrent S5 platforms (Thermo Fisher Scientific) following the manufacturer's protocols. OCCRA comprises 2031 unique DNA-based amplicons to detect SNVs, and CNVs, and 1701 RNA-based amplicons to detect unique fusion or structural variants[10]. The average read depth for the OCCRA panel was $5 \times 10^6$–$7 \times 10^6$ per sample for DNA and $1 \times 10^6$–$2 \times 10^6$ for RNA.

SNVs were retrieved with Ion Reporter software (version 5.2). Copy number measurements were retrieved with Ion Reporter software (version 5.2) for genes with >5 probes, including those that were validated for copy number gains as described elsewhere[10].

### St. Jude's Hospital (SJH) data curation
Data was downloaded from www.stjuderesearch.org/site/data/relapsed-all in December 2018. For our analysis, we included patients with Dx-R1 progression. To determine gene mutations as Dx unique, Shared, of Relapse unique, we included genes listed as "rise" as a shared mutation and genes listed as "fall" as Dx unique (Supplementary Data S3).

### Cytotoxicity analysis of variant-predicted drug response in paired Dx-R ALL samples
hTERT-immortalized mesenchymal stromal cells (MSCs) were provided by D. Campana (St. Jude's Hospital). hTERT-MSCs were seeded at 5000 cells per well in 200 µL of RPMI-1640 medium containing 10% fetal bovine serum (FBS, Invitrogen) and 1 µM hydrocortisone (Sigma) in a 96-well plate (Corning, Corning, New York, USA), 24 h prior to seeding with primary B-ALL or non-cancer bone marrow stem cells.

Primary samples were thawed at 37 °C for 1–2 min, washed 1× in warm RPMI-1640 medium containing 10% FBS and washed 2× with PBS, and stained with DAPI CFSE stain (Invitrogen) used to distinguish the ALL cells from the hTERT-MSC cells[45]. The media was removed before adding $5 \times 10^4$ B-ALL cells in 100 μL of AIM-V medium (Thermo Fisher Scientific).

Drug dilutions were prepared at 2× the final concentration (1 nM, 10 nM, 100 nM, 1 μM, 20 μM, and 30 μM) and 100 μl of each drug dilution was added to 100 μl of primary cells in each well. Cells were incubated with the drugs for 72 h at 37 °C in a 5% (v/v) $CO_2$ incubator. Drugs used in the study: Palbociclib, Trametinib, Ruxolitinib, and Vismodegib (Selleck Chemicals LLC, Houston, TX, USA). For PARP1/2 inhibitors, the drugs were prepared for final concentrations of 1 nm to 100 μM in 10-fold increments PJ34 (SelleckChem), 0.1 nM to 10 μM in 10-fold increments for Olaparib (SelleckChem).

After 72 h, CyQUANT Direct (Green) (Thermo Fisher Scientific) was added and incubated at room temperature for 1 hour. The plate was analyzed by a high content image analysis system (ImageXpress Micro XL). Images were taken using a 40 × 0.75 NA dry objective with the MetaXpress 5.0.2.0 software (Molecular Devices Inc) on the ImageXpress Micro XL epifluorescence microscope (Molecular Devices Inc). DAPI and GFP (green fluorescent protein) emissions were acquired simultaneously with a 505DCXR beam splitter (Dual-View; Optical Insights, LLC) with the optical filters for DAPI excitation or GFP emission, respectively. For the analysis of the proportion of living cells, images were taken once per site using 50-ms exposures, 2 × 2 binned resolution, with 100% of full lamp intensity for each channel, and 25 optical sections spaced 500 μm apart. Post-acquisition processing of images was performed using MetaXpress offline.

Viability was calculated by taking the mean of (DAPI and GFP double-positive (DP) cells)/ (DAPI single-positive cells) for each drug-treated well. To account for relative viability of the primary cells in the assay, the drug-treated viability was normalized to the calculated viability for the vehicle-treated (DMSO) cells. After 72 h in this platform, viability was generally >70% in vehicle-treated cells, but viability can vary between samples[44]. Thus, we set a viability of 45% as a minimum in vehicle-treated cultures. For PARP1/2 inhibitors, normalized viability was assessed by the summation of DP cells across four sites in drug-treated wells / summation of DP cells across 4 sites in DMSO-treated wells. IC50 concentrations were calculated in GraphPad PRISM version 9 (GraphPad Software, San Diego, CA, USA) with the method "log inhibitor concentration vs normalized response".

## Protein extraction and LC-MS/MS acquisition

Unless otherwise stated, reagents were purchased from Sigma Aldrich (St. Louis, Missouri, United States). Pellets of $0.5 \times 10^6$–$1 \times 10^6$ cells were lysed in 50 μl buffer containing 1% SDS (Fisher BioReagents, Pittsburgh, Pennsylvania, United States), 1X Pierce protease inhibitor (Thermo Fisher Scientific) in 50 mM HEPES (pH 8.0), followed by 5 min incubation at 95 °C and 5 min on ice. The sample was incubated with benzonase (EMD Millipore/Novagen, Massachusetts, USA) at 37 °C for 30 min to shear chromatin. Following benzonase treatment, each sample was reduced with 10 mM Dithiothreitol (DTT) dissolved in 50 mM HEPES pH 8.0 (37 °C, 30 min) and alkylated with 40 mM Chloroacetamide (CAA) dissolved in 50 mM HEPES pH 8.0 (30 min in the dark) and quenched in 40 mM DTT for 5 min at room temperature.

Lysates were cleaned using single-pot solid-phase-enhanced (SP3) bead technique[46] using hydrophilic and hydrophobic Sera-Mag Speed Beads (GE Life Sciences, Issaquah, Washington, United States). Proteins were bound to paramagnetic beads with 80% ethanol (v/v), incubated for 18 min at room temperature, and washed twice with 90% ethanol using magnetic isolation. Beads were then resuspended in 30 μl 200 mM HEPES, pH 8.0, and incubated with sequencing-grade trypsin (Promega Madison, Wisconsin, United States) at 1:50 protein ratio for 16 h at 37 °C, and afterwards acidified to pH 3–4 with formic acid. Peptide digests were de-salted on Nest Group Inc. C18 spin columns with 0.1% trifluoroacetic acid (TFA), eluted with 60% acetonitrile in 0.1% FA and dried in a vacuum concentrator. Dried samples were resuspended in 0.1% formic acid (FA).

## Library preparation–high pH reverse-phased fractionation (for BCCH cohort 1)

Depending on final protein amount, 1–4 μg of protein was taken from each sample and combined into one pool for fractionation. Fractionation was performed on a Kinetic EVO C18 column (2.1 mm × 150 mm, 1.7 μm core shell, 100 Å pore size, Phenomenex) connected to an Agilent 1100 HPLC system equipped with a diode array detector (254, 260, and 280 nm). A flow rate of 0.2 ml per minute was maintained on a 60 min gradient using mobile phase A (10 mM ammonium bicarbonate, pH 8, Fisher Scientific, cat. no. BP2413-500). Elution was with mobile phase B (acetonitrile, Sigma-Aldrich, cat. no. 34998-4 L) from 3 to 35%. Peptide fractions were collected each minute across the elution window. A total of 48 fractions were combined to a final set of 24 (e.g fraction 1 + 25 as final fraction 1), and dried in a SpeedVac centrifuge. Peptides were resuspended in 0.1% FA in water (SC235291, Thermo Scientific) prior to mass spectrometry analysis.

Peptides were analyzed using a Thermo Scientific Easy-spray PepMap™RSLC C18 column (75 μm × 50 cm, 2 μm, 100 Å; ES803), maintained at 50 °C on an Easy-nLC 1200 connected to a Q Exactive HF mass spectrometer (Thermo Scientific). Peptides were separated over a 3 h gradient consisting of Buffer A (0.1% FA in 2% acetonitrile) and 3–30% Buffer B (0.1% FA in 95% acetonitrile) at 250 nL/min. MS acquisition was performed with full scan settings between 400 and 1800 *m/z*, resolution of 60,000, AGC target of 5 e4, and Maximum IT of 75 ms. Stepped collision energy (NCE) was 28. MS2 scan settings were as follows: isolation window of 1.4 *m/z*, AGC target of 5 e4, maximum IT of 50 ms, at resolution of 15,000 and dynamic exclusion of 20.0 s.

## Library preparation–gas phase fractionation (for BCCH cohort 2&3)

Online gas-phase fractionation was performed. 1–2 μg de-salted peptides from select samples were combined into a single pool and analyzed in ten fractions, 1 μg per fraction, 3 h gradient. The first eight fractions (340–760 *m/z*) were analyzed over a 60 *m/z* window (i.e., 340–400 *m/z* is fraction 1) each with a loop count of 30 and window size of 2 *m/z*. The final two fractions (820–1180 *m/z*) were analyzed over 180 *m/z* window each, with a loop count of 30 and 6 *m/z* window.

Peptides from cohort 1 were separated on a Thermo Scientific Easy-spray PepMap™RSLC C18 column (75 μm × 50 cm, 2 μm, 100 Å; ES803) and from cohort 2&3 were separated on a PharmaFluidics 50 cm uPAC™ (ESI Source Solutions, Woburn, MA, United States), maintained at 50 °C on an Easy-nLC 1200 connected to a Q Exactive HF mass spectrometer (Thermo Fisher Scientific). The peptides were separated over a 3 h gradient consisting of Buffer A (0.1% FA in 2% acetonitrile) and 2–80% Buffer B (0.1% FA in 95% acetonitrile) at 300 nL/min. MS acquisition consisted of a MS1 scan ranges specified above for each fraction (AGC target of 3e6 or 60 ms injection time), and resolution of 120,000. DIA segment spectra were acquired with a AGC target 3e6, resolution 30,000, auto ms injection time, and stepped collision energy of 25.5, 27, 30. The library was also supplemented with DIA data from each individual sample.

## Sample MS acquisition

1 μg of peptides was injected for analysis for each sample. Samples were randomized and we obtained duplicate injections (cohorts 2&3) when possible. The DIA method consisted of a MS1 scan from 300 to 1650 *m/z* (AGC target of 3e6 or 60 ms injection time), and resolution of 120,000. DIA segment spectra were acquired with a 24-variable window format, (AGC target 3e6, resolution 30,000, auto for injection time), and stepped collision energy of 25.5, 27, 30. We added indexed

retention time (iRT) peptides (Biognosys, Schlieren, Switzerland) to each sample for retention time normalization and quality control.

## Full BCCH cohort–proteomics data analysis

The BCCH cohort consisted of three sample sets. The spectral library for each of the three sample sets was combined in Spectronaut into one library of 10,130 proteins. All DIA files were searched together with the combined spectral library. Briefly, the raw DIA files were analyzed with Spectronaut Pulsar X (Biognosys, Schlieren, Switzerland) using a human FASTA file from UniProt (reviewed 20200309). This FASTA file includes common contaminants. In addition, a FASTA file for iRT peptides, provided by Biognosys, was included in the search. Search was performed using the factory settings including specificity for Trypsin, Carbamidomethyl (C) as a fixed modification, and Acetyl (protein N-term) and Oxidation (M) as variable modifications. Precursor, and protein identifications false discovery rate (FDR) threshold was set to 1%, while the threshold for peptide was 0.5%. The data was normalized in Spectronaut based only on proteins identified in all samples and then further processed for batch effect removal by HarmonizR (version 0.0.0.9000)[47]. We utilized HarmonizR for batch effect correction is that it allows the retention of the proteins that otherwise would be dropped due to containing missing values (Supplementary Fig. S5).

For the analysis of the full cohort, data was filtered for proteins identified in at least 25% of each cohort, and the remaining missing values were imputed using a "down-shifted normal" imputation strategy resulting in a total of 7307 proteins used for analysis. For hierarchical clustering we filtered for highly variable proteins by calculating the relative FC (protein intensity/median protein intensity) for each protein, and selected only the proteins with a log2FC > 2 in at least 6 samples (3907 proteins).

The B-ALL analysis was restricted to proteins identified in all B-ALL samples (3857 proteins). Highly variable proteins with a log2FC > 1 in at least 5 samples were selected (935 proteins). For gene ontology analysis we used g:Profiler (BIIT! Research Group, g:Profiler version e104_eg51_p15_3922dba, database updated on 07/05/2021) with an FDR threshold of 5%. To attain better resolution of the pathway visualization, we followed a similar strategy previously described[48]; we limited the terms to GO: Biological Process (BP) and limited the number of intersections to 1000. The five terms with most significant adjusted p-value were selected for visualization.

## Survival analysis

Kaplan Meier survival analysis was performed in GraphPad Prism. Patients that had <5 year follow-up data were censored, meaning the sample was still included but the end of the follow-up data is indicated on the survival curve. A long rank test for trend was used to determine significance.

## Paired Dx-R cohort–proteomics data analysis

For a summary of data quality and filtering, refer to Supplementary Figs. S4, S5. The data was analyzed in Spectronaut as described above. A minimum of two peptides were required for quantitation. Protein intensities were normalized in Spectronaut using the "global" setting, which normalizes by median protein intensity per sample. Duplicate injections were averaged (mean). A pool of samples was created as a "standard" that was injected periodically throughout the course of the sample run, to measure data reproducibility and monitor MS performance. Proteins that were quantified with >50% CV in the 10 standards were removed from all samples as they are presumed to be un-reliably quantified. Furthermore, proteins identified in <10% of the samples were removed. Finally, missing values were imputed using a "down-shifted normal" imputation strategy. Briefly, a normal distribution was created out of the overall sample distribution, then shifted to lower values using magnitude of 3.5.

To summarize, we identified a total of 8153 unique proteins with an average of 6600 proteins per sample using a data independent acquisition approach (DIA) with a spectral library of 8183 proteins derived from gas-phase fractionated sample pool (Supplementary Fig. S6A). After quality assessment and data filtering (Supplementary Fig. 6B), we quantified an average of 5100 proteins per sample with at least two peptides at <0.05% FDR.

## Fluorescence-activated cell sorting to isolate mature B-cells

To investigate comparison to B-cells, we included data from naïve and memory B-cells isolated by flow cytometry from age-matched PBMCs. To decrease processing and patient-specific variability and increase sorting efficiency, we combined PBMCs from five patients that did not have any hematological malignancies and were in a similar age range (see Supplementary Data S1). Briefly, we thawed 1 vial ($-5 \times 10^6$–$10 \times 10^6$ cells per vial) and washed once with sterile FACS buffer (PBS (Thermo Fisher Scientific) + 2% FBS (Thermo Fisher Scientific)). We used $25 \times 10^6$ live cells for staining. Cells were pelleted and resuspended in approximately 100 μl of FACs buffer. We utilized a 8-color flow panel designed to identify CD45+ lymphocytes and then optimally separate T-cells and B-cells and their respective subpopulations; for this experiment we aimed to isolate two different mature B-cell populations, naïve B-cells (CD45 + CD3-CD19 + CD10-CD20 + CD27−). and memory B-cells (CD45 + CD3-CD19 + CD10-CD20 + CD27+). The panel consists of CD45-AF488 (HI30, 1:14)), CD3-BV510 (UCHT1, 1:10), CD19-APCFire (HIB19, 1:5), CD4-AlexaFluor700 (SK3, 1:10), CD8-PECy7 (RPA-T8, 1:10), CD10-BV421 (HI10a, 1:8), CD20-APC (2H7, 1:14), CD27-PE (O323, 1:10) (BioLegend). We added BV staining cocktail for optimal performance of the BV dye and human FC blocker to reduce unspecific binding. Staining was performed for 20 min at 4 °C in the dark. After staining, the cells were washed with FACS buffer, centrifuged at 1500 rpm and resuspended in 1 mL of FACS buffer. Just prior to sorting, we added 7AAD (1:3) for viability. Sorting was done at the Center for Molecular Medicine and Therapeutics flow sorting core at BCCHR on an Astrios FACS sorter.

## Statistical analysis

We developed a hybrid method for comparing quantified proteins to assess protein stability. Our method employs equivalence and differential expression testing to create a robust comparison between statistically different and equivalent proteins for a given pair of samples.

First, we normalized the quantitative protein data by median centering and removed samples 1, 2, 3, and 9. Next, we used coefficient of variation (CV) based filtration to filter out proteins with unstable quantification. We removed proteins with >20% CV between the technical replicates for each sample. Only proteins that passed this threshold in each pairwise comparison remained, resulting in all values and imputed data.

We conducted statistical tests on each pair of sample comparisons and their two technical replicates. Differential expression testing was performed using an independent two-sample t-test with the ttest_ind function from the stats package. Equivalence testing was done with two one-sided t-tests using the TOST two_raw function from the TOSTER package. We set logFC < −1 and logFC > 1 as boundaries for equivalence. Proteins for a given pair are run through both tests, and both p-value sets are corrected using the pvalue.adjust function's FDR method with a cutoff of 0.05. To identify which protein is statistically different, equivalent, or unexplained, we look at the logFCs. If the logFC for a given protein is outside of the −1, 1 boundary, then the adjusted p-value from the different test is considered. If the logFC is inside the −1, 1 boundary, the adjusted p-value from the equivalence test is considered. The protein is labeled as statistically different or equivalent if the corresponding adjusted p-values are lower than the 0.05 cutoff. Proteins with higher than 0.05 adjusted p-value are labeled as statistically unexplained.

To obtain a list of equivalent proteins for gene ontology enrichment, we first ensured the protein was equivalent in at least two of the seven Dx-R pairings. We next removed the proteins attributed to "housekeeping" functions by creating a cancer vs. non-cancer pairings equivalent list, which also required the protein was detected in at least two pairings. Housekeeping proteins were then removed from our paired Dx-R equivalence list. To obtain the "difference" list, we compiled all proteins that were deemed significantly different in each Dx-R pairing (BALL01 was excluded since this patient does not contain a Dx).

Gene ontology enrichment analyses was performed as described above. For visualization of the enriched terms, we employed Cytoscape[49] (version 3.8.2) and utilized the "EnrichmentMap" and "AutoAnnotate" packages. The background list for all GO analyses was a list of all proteins quantified in the data set.

### Identification of cancer-associated proteins
Proteins were selected from two large-scale pediatric cancer studies that identified commonly mutated genes, genes of prognostic importance, and potential cancer drivers[9,10] (Supplementary Data S6). In addition, targets were included from the OCCRA panel, which is similar to the Oncomine panels used in NCI-COG Pediatric MATCH precision medicine trial[21] (Supplementary Data S6).

For statistical analysis, the "LIMMA" package for R (version 3.52.2) was used[50]. P-value was adjusted using the Benjamin-Hochberg and the FDR threshold was 5%. Circos plots were created in R using the package "Circularize" (version 0.4.15); data was first filtered for proteins that were overexpressed relative to the non-cancer controls. Pearson correlation coefficient was calculated for each pairing based on proteins that were overexpressed in Dx.

### RNA sequencing
BCCH Cohort: RNA was extracted from cell pellets of $1 \times 10^6$ cells using the RNeasy Mini kit (QIAGEN)) and the concentration of RNA was determined using Qubit 3.0 Fluorometer (Invitrogen). RNA quality was measured by bioanalyzer (Agilent). Library preparation from these RNA samples was performed on the Ion Chef and Ion Torrent S5 platforms using the Ion AmpliSeq™ Transcriptome Human Gene Expression Panel, Chef-Ready Kit (A31446, Thermo Fisher Scientific) following Ion AmpliSeq™ Library Preparation on the Ion Chef™ System Quick Reference. Each sample was prepped and sequenced in duplicate. The resulting cDNA library was quantified using Ion Library TagMan™ Quantification Kit (Thermo Fisher Scientific). Targeted sequencing was performed on the Ion Chef and Ion Torrent S5 platforms (Thermo Fisher Scientific) following manufacturer's protocols (TFS). The Ion AmpliSeq Transcriptome Human Gene Expression Assays measure gene expression of over 20,000 RefSeq genes in a single assay simultaneously. An average of nine million reads per sample was obtained. Data processing and quality control was performed using the AmpliSeqRNA plug-in for Ion Torrent S5.

### RNA-sequencing analysis
BCCH Cohort: All data was normalized by reads per million for each sample. The data was also scaled by $log2(x+1)$. The data was filtered for transcripts that were measured in all samples, then duplicate measurements were combined by mean for each transcript. Pearson correlations were performed using the "pheatmap" package in R (version 1.0.12) and the T1vsT2 scatter plots were plotted with ggplot2 using stat_cor(method=pearson).

TARGET Cohort[16]: The data was downloaded from the Treehouse Childhood Cancer Initiative at the UC Santa Cruz Genomics Institute. Only paired Dx and R pediatric ALL samples were selected for analysis. The data was already normalized by parts per million and scaled by $log2(x+1)$. Pearson correlations were performed as described above for the BCCH cohort.

### Primary cell irradiation
hTERT-MSCs were seeded at 70% confluency per well in 4 mL of RPMI-1640 medium containing 10% fetal bovine serum (FBS, Invitrogen) and 1 μM hydrocortisone (Sigma) 24 h prior to seeding with primary B-ALL or stem cells from bone marrow. To seed primary cells, RPMI-1640 complete medium was removed before adding $3.2 \times 10^6$ primary cells, recovered from cryopreserved samples, in 4 mL of AIM-V medium. Both primary cells and hTERT-MSCs were incubated at 37 °C in a 5% (v/v) $CO_2$ incubator. After 24 h co-culture, primary cells were removed from co-culture and treated with 1 Gy X-irradiation or sham conditions. Primary cells were then added back to hTERT-MSC co-culture. Half of the primary cells were fixed 0.5hrs after irradiation, with the other half fixed 24 h after irradiation. Cells were concentrated onto a slide using the Epridia Cytospin 4 centrifuge (Fisher) and fixed in methanol at −20 °C for 5 min before storage at −20 °C.

### Primary cell immunofluorescence
Cells were concentrated onto slides using the Epridia Cytospin 4 centrifuge (Fisher) and fixed in methanol at −20 °C for 5 min. Cells were outline with a PAP pen (abcam) and blocked in PBS with 0.2% Triton X-100 and 3% BSA for 1 h at room temperature. Antibodies were diluted in PBS with 0.2% Triton X-100 and 3% BSA. Primary antibodies were diluted (γH2Ax 1:500, PARP1 1:250) and incubated with slides overnight at 4 °C. Cells were then washed three times in PBS. The slides were incubated with diluted secondary antibodies (Alexa Fluor 1:2000) at room temperature for 1 h in the dark. Slides were washed three times in PBS and incubated with Hoechst stain for 15 min. Slides were then washed two times in PBS and coverslips were mounted with ProLong Gold Antifade (Invitrogen) reagent.

### Confocal microscopy and image analysis
Fixed cells were imaged using the Fluoview software (Olympus) connected to the Olympus Fluoview FV10i confocal microscope. Image stacks of 5 optical sections with a spacing of 0.5 μm through the cell volume were taken using a $60 \times 1.2$ NA oil objective. PARP1 stained with AlexaFluor 594 was imaged at 50% sensitivity, and 40% laser power. γH2Ax stained with Alexa Fluor 647 was imaged at 50% sensitivity, and 40% laser power. Hoechst nuclear stain was imaged at 40% sensitivity and 13% laser power. ImageJ v1.46j (National Institute of Health) was used to generate maximum intensity Z-projection of the fluorescent channels, and subsequent analysis. Nuclear masks were generated for each cell (Make Binary) and the resulting region of interest (Analyze Particles) was used to identify the nuclear region of analysis for γH2Ax and PARP1 channels. γH2Ax foci per cell was quantified using the Find Maxima process (prominence > 750). PARP1 nuclear fluorescence was quantified using the Measure analysis. Statistical analysis was performed using GraphPad Prism with Welch's t-test, as indicated in each figure. The results were considered statistically significant at $P < 0.05$.

### Reporting summary
Further information on research design is available in the Nature Portfolio Reporting Summary linked to this article.

## Data availability
The raw MS data generated in this study have been deposited to the Proteome Consortium (http://www.proteomexchange.org) via the MassIVE (https://massive.ucsd.edu/) partner repository data set MSV000091012. The raw OCCRA targeted DNA and RNA-fusion sequencing data generated in this study have been deposited to the National Center for Biotechnology information (NCBI) Sequence Research Archive (SRA) (https://www.ncbi.nlm.nih.gov/sra) with the project ID PRJNA985851 and the raw OCCRA RNA-sequencing data generated in this study have been deposited to the NCBI SRA (https://www.ncbi.nlm.nih.gov/sra) with the project ID PRJNA985381. Source data are provided with this paper. The publicly available WES data from

69 pediatric ALL patients from SJH used for the genomics analyses is available in the SJH cloud and can be accessed individually with the patient IDs listed in Supplementary Data 3. The publicly available RNA sequencing data for the 35 pediatric ALL patients from the TARGET cohort was downloaded from the Treehouse Childhood Cancer Initiative at the UC Santa Cruz Genomics Institute, from the data set titled "Tumor cohorts for Vaske et al. publication (October 2019)". Source data are provided with this paper. The remaining data are available within the Article, Supplementary Information or Source Data file. Source data are provided with this paper.

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

## Acknowledgements

This work was supported by the Michael Cuccione Foundation (MCF) and the BC Children's Hospital Foundation through the Better Responses through Avatars and Evidence (BRAVE) Initiative. Salary support was provided by the MCF (C.J.L., G.S.D.R., C.A.M., P.F.L.), the Canada Research Chairs Program (CRC-RS 950-230867, P.F.L.), the Canadian Institutes of Health Research (C.A.M., P.F.L.), the Michael Smith Foundation for Health Research Scholar Program (16442, P.F.L.) and the University of British Columbia (A.L., E.K.E.). We thank Pascal Leclair for contributions made through the BRAVE Initiative and technical assistance. We gratefully acknowledge the participation of the patients and families that made this study possible and the BC Children's Hospital nurses and physicians and Biobank staff for their tremendous efforts in collecting and maintaining specimens. We also acknowledge our consultations with the Modeling node and Proteomics node of the Precision Oncology for Young People Translational Research Program.

## Author contributions

A.C.L. carried out the targeted NGS and proteomics experiments and corresponding analyses. A.C.L. performed the targeted drug response experiments with assistance from N.M.A. PARP and γH2Ax characterization and PARP inhibitor response experiments and analysis were carried out by M.G. and J.R. E.K.E. assisted with the proteomics data analysis. C.A.M. and P.F.L. supervised the project and A.C.L., C.A.M., and P.F.L. wrote the manuscript with contributions from N.R., C.J.L., and G.S.D.R.

## Competing interests

The authors declare no competing interests.
