## [Peer Review File · Nature Communications]

Targetable lesions and proteomes predict therapy sensitivity through disease evolution in pediatric acute lymphoblastic leukemiaReviewers' Comments:

Reviewer #1:

Remarks to the Author:

The manuscript describes an interesting study of the maintenance of targetable lesions and associated features in pediatric ALL. Specifically, the investigators analyzed retrospective pairs of ALL samples at the diagnosis and relapse stages via genomic and proteomic data. Such matched samples are precious and, while statistical power is limited by the numbers of samples, their main finding of high rates of durability of actionable genomic mutations and stable proteomic features over the course of the disease would seem to have important medical implications. This is an extremely timely topic of interest for readers and certainly within the journal's scope. There are a few technical questions for the authors.

1. Figure panel 3A seems confusing on a few counts. In the caption, the authors mention using the two-one-sided t-test, but it is not clear as to what the advantages are over standard 2-sided t-testing of an equal null and alternative hypotheses. The caption concludes with saying that the P-value is actually calculated with the Wilcoxon rank test. Please clarify between t-test and Wilcoxon.
2. Also with respect to panel 3A, the investigators seem to be describing a fairly large number of statistical tests. For example, just the 2 top categories appear to encompass $2 * 6 * 6 = 72$ tests. I did not see any obvious multiple test correction for this part.
3. Is LIMMA mentioned on line 932 from BioConductor? Please include reference.
4. Some figure details are not easy to make out, for example GO-codes in Fig 2C and axes in Fig 3C.

Reviewer #2:

Remarks to the Author:

Lorentzian and colleagues provided a detailed genome and proteome analysis of paired diagnostic and relapsed pediatric ALL samples. This study supports the notion that proteomes of the malignant cells in B-ALL are stable during the disease and provides a proteomic analysis-based pipeline for the potential therapeutic targets discovery. Importantly, authors also experimentally demonstrated poor selectivity towards targeted therapies (e.g CDK4i, MEKi and more) based on the genetic analysis alone.

In my view, this work delivers significant technological and bioinformatics accomplishments for the field of personalized medicine and pediatric oncology.

The work fits well with the established criteria in the literature of precision medicine based on proteogenomic approaches.

Although the work is original in terms of methodology, design, and disease type, the discovery potential of this approach was validated on PARP1, which is a well-known therapeutic target in this type of leukemia. In fact, the utility of PARP targeting in ALL was previously demonstrated in several studies (Esposito et. al , Nature Med 2015 Dec;21(12):1481-90.doi: 10.1038/nm.3993., Piao et al, PMID: 27894958 DOI: 10.1016/j.canlet.2016.11.021) and highlighted in recent reviews (e.g. Padella et al., Journal of Hematology & Oncology (2022) 15:10 <https://doi.org/10.1186/s13045-022-01228-0>). This literature deserves to be cited.

Although authors decided to prioritize PARP in their validation studies, this somewhat diminishes the innovation potential of their findings.

One way to potentially strengthen their claim might be validation in their well-annotated samples whether reported determinants of PARP resistance in ALL (e.g. MLL subgroup, (Esposito et. al , Nature Med 2015) indeed hold true.

Major comments:

- 1) Authors claim that the set of targetable lesions persists in pediatric ALL samples they profiled. However, in their validation set, e.g. Figure 4G, no matched diagnosis relapsed samples were tested for their sensitivity to Olaparib or PJ34. Formally, it is possible that sensitivity to this therapy evolves in the matched patients despite similar protein expression level.
- 2) More detailed discussion of the additional possible targets are welcomed.

Minor comments:

- 1) Lane 454, what is GFP viability dye?
- 2) Figure 1G, the coloring does not fit the legend.

Reviewer #3:

Remarks to the Author:

Lorentzian et al present a comprehensive retrospective analysis of paired ALL diagnosis and relapsed specimen. They argue that a major challenge for prospective precision oncology approaches is a limited understanding of the persistence or evolution of targetable lesions and their associated proteins or pathways, and responses to targeted agents, that may be gained or lost at relapse. Consequently, they suggest that because proteins are the actual therapeutic targets for most oncology drugs, it is imperative to understand the pediatric tumor proteome to determine how the response to therapy may change through progression.

I applaud the authors for their effort to interrogate the proteome in diagnostic and relapsed ALL samples, and I found some of their findings intriguing. However, I have a fundamental disagreement with what I believe is a central premise in this work. From my experience in analyzing both RNA and protein expression experiment, the number of differentially expressed genes or proteins induced by therapeutic intervention is relatively small compared to the number of genes that vary due to cell identity. In PCA analysis, the first principal component often discriminates between different cell models – meaning, variation due to treatment is relatively small in magnitude. So, how much should we expect the proteome to change from diagnosis to relapse? Most of the proteins—including cancer associated proteins—are probably not consequential to a specific therapeutic response. In fact, for targeted therapy, a single mutation can disrupt drug binding and cause resistance to therapy, and this change could be difficult to detect from proteomic data alone. In the Ma et al paper cited by the authors, while there was considerable overlap in variants between diagnosis and relapse, six pathways were frequently mutated, with NT5C2, CREBBP, WHSC1, TP53, USH2A, NRAS and IKZF1 mutations enriched at relapse. In fact, relapse-specific mutations in NT5C2 were relatively common, and it is known that mutations in this gene confer resistance to certain ALL treatments. Therefore, the distinction between diagnosis and relapse could be subtle. It would be helpful for the authors to acknowledge these findings from the Ma et al paper. Furthermore, are there data sets comparing gene expression between diagnostic and relapsed ALL tumors? Are the number of genes that change small as in this study?

The authors then go on to identify PARP1 as a protein that is over-expressed in B-ALL compared to non-cancer bone marrow-derived stem cells, and whose expression remains consistently elevated in diagnosis and relapse. However, PARP1 plays a role in the cell cycle and its expression might be higher in cycling cells (see PMID 24367566 and 29886395). Thus, is the increase in PARP1 expression in B-ALL simply a reflection of a higher proliferation rate? Perhaps this could explain why in Figure 4F, B-ALL cells had higher levels of nuclear PARP1 compared to BMSC independent of genotoxic stress at the 24 h timepoint (there seems to be a bimodal distribution as well – perhaps cell cycle related?). Likewise, while PARP inhibitors appeared to be more effective in B-ALL cells compared to BMSC cells, could this simply be a function of higher growth rate, as cycling cells are more prone to DNA damage and PARP1 is important for the DNA damage response? One possible way to address the effect of

proliferation would be to repeat the experiment in Figure 4H with Topo1 inhibitors, which are also selective for proliferating cells, and anti-mitotic agents like vincristine. Finding a large therapeutic window for PARP inhibitors, but not the other agents, would strongly support the author's rationale to target PARP1 in B-ALL.

While I agree the proteome data set is interesting and worth reporting, these points above dampen my enthusiasm for the paper. Moreover, it would be helpful to clarify the following points for the reader:

In Figure 1B, it is difficult to interpret the Circos plots. Perhaps it would be better to represent this data in a heatmap with variants depicted as circles whose radius indicates frequency of occurrence and with color representing degree the variant is shared in diagnosis and relapse?

For Figures 1B-1E, what is the set of variants analyzed? Is this the set of all somatic mutations identified across all samples? Or somatic mutations reported as cancer drivers?

In Figure 1C, wouldn't one matched pair be represented as a point in each of the three groups (Dx, R, and Share), with the 3 points summing to 1.0? The data is plotted in a way that loses the relationship between matched sample pairs. Please consider presenting the data as a stacked barplot (as is done in the supplement): this would more clearly show the variation in shared vs unique variants found per sample.

In Figure 1D-E, the relationship between % shared and tumor subtype is striking. It would be helpful to confirm that these differences are statistically significant using the appropriate tests (maybe ANOVA?).

Figure 1F is a nice way of showing variation between diagnosis and relapse. It is striking that for each target, there are examples of variation between diagnosis and relapse. This is most evident for the CDK4/6, MEK, and MTOR pathways. Is the nature of variants different in those that are consistent vs. those that are variable?

In Figure 1G, I would suggest representing EC50 on a log scale, especially for the MEK plot where EC50 values range at least two orders of magnitude. It likely that this would make the difference between target positive and negative groups in the MEK plot look much smaller. The plot in H, which is on the log scale, clearly indicates close agreement between diagnosis and relapse.

In Figure 2C, the authors conclude that proteome analysis shows stability through progression. However, there were only 6 matched samples in this analysis, and 2 of the 6 clustered diagnosis and relapse in different groups. This conclusion is not supported by the data.

The survival analysis in Figures 2D-E is interesting. There were no events in P2, P3, and P7 – is this correct? P4 was the cluster showing the poorest outcome, but this cluster was also enriched for BCP-ALL and HR tumors. Would the analysis of the clusters P2, P3, P4, and P7 based on risk group (not cluster group) result in a significant difference in outcome? Is the result largely driven by BCL-ALL?

It's difficult for me to follow the statistical analysis performed in Figure 3B. Did the authors look at all significant differentially expressed proteins between matched diagnosis and relapse, and identify those which are consistently different (meaning enriched in diagnosis or enriched in relapse across all samples)? Only equivalent proteome and relapsed-enriched proteome is presented. Shouldn't there also be a diagnosis-enriched proteome? Would GSEA on the non-equivalent proteome, restricted to proteins that change with large logFC in the same direction for diagnosis and relapse, be informative?

In Figure 3F, while it is true that there is significant correlation between diagnosis and proteome, in six of the seven cases, there are clear outliers. What are these? Perhaps they have biological significance?

Reviewer #4:

Remarks to the Author:

In the manuscript titled "Persistence of targetable lesions, predicted therapy sensitivity and proteomes through disease evolution in pediatric acute lymphoblastic leukemia" authors applied genomics and proteomics to examine, proteogenomic evolution in relapsed pediatric acute lymphoblastic leukemia patients. In my reviewing, the manuscript in its current form is not suitable for publication in Nature communication.

Major concern:

1. The manuscript is not coherent, at times it is difficult to follow the correlation between different sections of results. The manuscript lacks a clear finding and novelty. The rationale for using different proteomics analysis presented in the manuscript is not clearly defined. It is also not clear how genomics and proteomics are compared, whether or not same patient cohort was used for proteomics and genomics analysis. Overall manuscript lacks clarity, flow of the story and novel findings.

2. Authors used different cohorts for different analysis, and it is not clear how each cohort is related to each other and how they were compared between each other.

For example, in section "Next-generation sequencing (NGS) reveals stability of affected genes through ALL disease progression" authors writes, "we sourced 71 bone marrow biopsies (n=44 ALL at initial diagnosis (Dx), n=21 ALL at relapse (R), n=6 non-cancer bone marrow) from pediatric patients seen at BC Children's Hospital (BCCH) (Supplementary Table S1) and publicly available whole-exome sequencing (WES) data from 138 specimens collected and analyzed by the St. Jude's Children's Research Hospital (SJH)" it is not clear how the 71 bone marrow biopsies data and 138 whole-exome sequencing data is used/compared?

Again, in line 97 author writes "We first explored the mutational landscape in paired progression samples (n=10 B-ALL, n=1 T ALL) via targeted, pediatric cancer-focused NGS analysis (10)." It is not clear whether samples in this cohort are part of which cohort or is it a separate cohort?

Again, in line 107 author writes, "To determine the generalizability of this finding, we mined public NGS datasets from ALL cases (n=49 B-ALL; n=20 T-ALL) treated at St. Jude's Hospital (SJH)." It is not clear whether samples in this cohort are part of which cohort or is it a separate cohort?

Again, in line 125 author writes "In the SJH and BCCH cohorts (n=80 paired samples)" it is not clear which 80 patients author is referring to?

Again, in section "Global proteome analysis shows stability through progression and groups cases with poor outcome" in line 153 author writes "we next conducted a comprehensive analysis of 48 primary (n=39 B-ALL, n=9 T-ALL) specimens from Dx and R, including 14 specimens from 6 patients with matched biopsies taken at diagnostic and subsequent relapse timepoints. It is not clear whether samples in this cohort are part of which cohort or is it a separate cohort?

3. In line 165, author writes "Samples clustered distinctly by B-ALL, T-ALL and non-cancer monocytes and cell lines (Fig. 3B)." However, Fig. 3B in the manuscript represents pathway enrichment analysis and not cluster analysis. Please correct it. Also, I would suggest adding a PCA plot based on all quantified proteins in each group to show the separation of different groups of samples. I would also suggest author to provide a PCA cluster analysis based on age and sex of the patients to show the effect of age and sex if any.

4. In Figure 2B and 2C, it is not clear what the fold change refers to, meaning how the fold change is calculated between different sample type/cytogenetic group? Also, for different groups there are different numbers of samples, and different n could have an impact on the statistical analysis. I would like authors to comment if they evaluated the effect of size on fold -change significance.

5. In Fig. 4A it is not clear why a random cutoff for log₁₀ intensity at 6 is selected. There are many other proteins which have higher fold change and higher intensity than PARP1 and those proteins are not discussed.

Point-by-Point Responses to Reviewers

Summary of main revision experiments:

1. NEW investigation of stability at the mRNA expression level in matched B-ALL patient

samples: We performed NEW whole transcriptome analysis on 4 matched pediatric ALL diagnosis (Dx) and relapse (R) progression pairs sourced from the BC Children's Hospital biobank. Three of the four pairs clustered together based on Pearson correlation. The one pair that did not also had the lowest genome and proteome stability. To increase the power of this study, we analyzed a whole transcriptome data set of 35 matched pediatric ALL progression pairs (70 samples) sourced from the TARGET initiative (Vaske et al, JAMA 2019). In this new data, hierarchical clustering based on Pearson correlation of all samples showed 34% of the pairs clustered as neighbors, while an additional 17% of pairs were within the same major cluster (new Supplementary Figure S11B).

For each pair Dx vs R, Pearson correlation of cancer associated gene products also demonstrated high stability with 83% of pairs having an r value $>.85$ on the transcript level (new Supplementary Figure S11C), consistent with our protein level analysis (Figure 3F). In the BCCH cohort, two pairs that demonstrated very high stability at the genome and global proteome and CAP proteome level also demonstrated the highest stability at the transcript level (BALL01 $r=0.94$, BALL04, $r=0.98$), while the patient that had the lowest stability overall also had the lowest CAP stability (BALL06 $r=0.7$). These transcriptome findings confirm a similar pattern of stability between Dx and R (or R-R) that we observe with the genome and proteome.

2. NEW investigation of cell proliferation and cell cycle markers that may be related to

sensitivity to PARP1 inhibitors: We performed immunofluorescence analysis of pHisH3 in primary B-ALL samples and B-ALL cell lines and found very low levels of dividing cells (new Supplementary Figure S16A,B). And, we directly assessed the expression of 14 cell cycle proteins previously found to peak in abundance in G2 and M relative to G1 phases via proteomics of intracellular immunolabelled cell subsets analysis (Ly et al, eLife 2017). While PARP1 expression was uniformly elevated in Dx and R samples, relative to non-cancer bone marrow samples, none of the 14 cell cycle regulated proteins showed a similar elevation, and none of these proteins correlated to PARP1 levels (new Supplementary Figure S16C).

3. NEW analysis using FDR correction for equivalence test: we performed a multiple testing correction, and adjusted p-values for the false discovery rate (FDR) with a cut-off of 5% (Figure 3A-B) (Supplementary Figure S9) and (NEW Supplementary Figure S10).

4. NEW analysis of PCA plots: We conducted PCA analysis to demonstrate the separation of samples based on sample type and the results are presented in NEW supplementary Figure S7. Based on PC1 and PC2, the samples separate by ALL, non-cancer BM and cell line (Supplementary Figure S7A). PC3 and PC4 further separate the ALL group into B-ALL and T-ALL (Supplementary Figure 7B). The symbols represent sex to demonstrate there are no obvious biases based on sex. Additionally, we investigated the contribution of each metavariable per PC using the Kendall correlation method and produced a correlation heatmap (Supplementary Figure S7C). We conclude that age and sex do not contribute significantly to the variation in the data and that the variation is primarily driven by sample type (B-ALL, T-ALL, Non-Cancer BM, or Cell Line).

Reviewer #1, expertise in proteogenomics (Remarks to the Author):

The manuscript describes an interesting study of the maintenance of targetable lesions and associated features in pediatric ALL. Specifically, the investigators analyzed retrospective pairs of ALL samples at the diagnosis and relapse stages via genomic and proteomic data. Such matched samples are precious and, while statistical power is limited by the number of samples, their main finding of high rates of durability of actionable genomic mutations and stable proteomic features over the course of the disease would seem to have important medical implications. This is an extremely timely topic of interest for readers and certainly within the journal's scope. There are a few technical questions for the authors.

Response. We are thankful for the reviewer's comment that the manuscript examines a timely topic, and that our findings are likely to have important medical implications and fall within the scope of *Nature Communications*. We address each of the few technical questions in the following paragraphs.

1. Figure panel 3A seems confusing on a few counts. In the caption, the authors mention using the two-one-sided t-test, but it is not clear as to what the advantages are over standard 2-sided t-testing of an equal null and alternative hypotheses. The caption concludes with saying that the P-value is actually calculated with the Wilcoxon rank test. Please clarify between t-test and Wilcoxon.

Response. We agree with the reviewer that our original submission was not clear on these points. We used a t-test for difference and a two-one-sided t-test for equivalence to measure the statistical difference and equivalence between proteins within pairings, as the assumptions for t-tests were valid with the data. We used the Mann-Whitney U test to compare the two groups (same patient-different timepoint vs different patient-same timepoint). We used a non-parametric test to highlight the differences between the two groups on the boxplot because the groups did not follow a normal distribution. We revised the figure legend and the methods to better clarify the statistical tests that we used for each analysis.

The modified figure legend now reads:

“Summary of tests for equivalence (Two-one-sided t-test (TOST) for equivalence, boundaries between $\log_2FC < -1$ and $\log_2FC > 1$, $FDR < 5\%$) of protein abundance between different groups and pairings. Only statistically measurable proteins are represented. Each dot represents the mean equivalence or difference of all protein abundance for a pairing. To highlight the difference between the two groups' percent equivalence boxplots (same patient-different timepoint vs different patient-same timepoint), the Mann-Whitney U test was performed on the two groups, *** indicates $p\text{-value} \leq 0.001$.”

2. Also with respect to panel 3A, the investigators seem to be describing a fairly large number of statistical tests. For example, just the 2 top categories appear to encompass $2 * 6 * 6 = 72$ tests. I did not see any obvious multiple test correction for this part.

Response. In our revised manuscript, we now include multiple testing correction for the proteins tested within a pair and adjust p-values for the false discovery rate (FDR) with a

cut-off of 5%. We do not perform any tests across pairs or categories and hence correction at this level is not required and possible.

The modified text and figure legends now clearly state the type of multiple test correction used as follows:

Figure legend: “Summary of tests for equivalence (Two-one-sided t-test (TOST) for equivalence, boundaries between $\log_2FC < -1$ and $\log_2FC > 1$, $FDR < 5\%$) of protein abundance between different groups and pairings. Only statistically measurable proteins are represented. Each dot represents the mean equivalence or difference of all protein abundance for a pairing. To highlight the difference between the two groups' percent equivalence boxplots (same patient-different timepoint vs different patient-same timepoint), the Mann-Whitney U test was performed on the two groups, *** indicates $p\text{-value} \leq 0.001$.”

Text: “To better understand inter- and intra- patient stability among the disease states, we tested proteins for statistically significant equivalence in all possible patient and timepoint pairings by two-one-sided t-test (TOST) and corrected for a 5% false discovery rate (FDR).”

3. Is LIMMA mentioned on line 932 from BioConductor? Please include reference.

Response. In our revised manuscript, we include a reference on line 708 for Bioconductor – Ritchie ME et al. (2015) *Nucleic Acids Research* 43(7) e47

4. Some figure details are not easy to make out, for example GO-codes in Fig 2C and axes in Fig 3C.

Response. We agree with the reviewer. Thus, in our revised version, we have altered the Figures so that no Font is less than 5-point size.

Reviewer #2, expertise in DNA damage and pediatric ALL (Remarks to the Author):

Lorentzian and colleagues provided a detailed genome and proteome analysis of paired diagnostic and relapsed pediatric ALL samples. This study supports the notion that proteomes of the malignant cells in B-ALL are stable during the disease and provides a proteomic analysis-based pipeline for the potential therapeutic targets discovery.

Importantly, authors also experimentally demonstrated poor selectivity towards targeted therapies (e.g CDK4i, MEKi and more) based on the genetic analysis alone.

In my view, this work delivers significant technological and bioinformatics accomplishments for the field of personalized medicine and pediatric oncology.

The work fits well with the established criteria in the literature of precision medicine based on proteogenomic approaches.

Response. We are thankful for the reviewer’s comment that the manuscript delivers significant technological and bioinformatics accomplishments for the field of personalize

precision oncology. We address each of the reviewer's comments in the following paragraphs.

1. Although the work is original in terms of methodology, design, and disease type, the discovery potential of this approach was validated on PARP1, which is a well-known therapeutic target in this type of leukemia. In fact, the utility of PARP targeting in ALL was previously demonstrated in several studies (Esposito et al., *Nature Med* 2015 Dec;21(12):1481-90.doi: 10.1038/nm.3993., Piao et al, PMID: 27894958 DOI: 10.1016/j.canlet.2016.11.021) and highlighted in recent reviews (e.g. Padella et al., *Journal of Hematology & Oncology* (2022) 15:10 <https://doi.org/10.1186/s13045-022-01228-0>). This literature deserves to be cited.

Response. We agree with the reviewer's comment. And, we are grateful to the reviewer for highlighting each of these omissions. In the revised manuscript, we have included each of these references where appropriate.

2. Although authors decided to prioritize PARP in their validation studies, this somewhat diminishes the innovation potential of their findings. One way to potentially strengthen their claim might be validation in their well-annotated samples whether reported determinants of PARP resistance in ALL (e.g. MLL subgroup, (Esposito et al., *Nature Med* 2015) indeed hold true.

Response. We thank the reviewer for raising this important point. Esposito et al. found gamma-H2Ax foci were elevated in primary mouse hematopoietic cells transformed with AML-associated transcription factors (AML-ETO, PML-RAR α , or MLL-AF9) indicating ongoing DNA damage or replication stress. We report a similar elevation in primary B-ALL patient samples (Fig. 4E). Further, Esposito et al. found reduced expression of BRCA2 and RAD51 within whole-cell lysates from AML1-ETO-transformed cells and PML-RAR α -transformed cells relative to MLL-AF9-transformed cells. Mechanistically, Esposito et al. show the induction of HOXA9 by MLL fusions reduces PARP sensitivity.

To investigate this pathway, we measured gamma-H2Ax foci in primary B-ALL samples after 24 hours in culture as a measure of replicative stress, similar to Esposito et al. We found elevated levels of gamma-H2Ax foci, indicative of DNA damage, which were correlated with a lower IC50 value for both Olaparib and PJ34 in the three B-ALL samples and two bone marrow stromal cell samples that we tested (Reviewer Figure 1A).

A. Select samples were separately treated with PARP inhibitors to obtain inhibitory concentration 50 (IC50) values or cultured for 24 hours, fixed and stained for γ H2Ax foci, as a measure of DNA damage induced by replicative stress. IC50 values were correlated to average γ H2Ax foci per cell per sample. Pearson's correlation coefficients (r) are displayed for each inhibitor plotted on the graph.

B. Box plots representing the protein abundance of proteins that are potential determinants of sensitivity to PARP inhibitors, PARP1 and RAD51, for Non-cancer bone marrow (BM) (red), diagnosis (Dx) specimens (tan), and relapse (R) specimens (black).

Next, we examined the expression of markers of PARP sensitivity or resistance in our proteomics dataset, including RAD51, BRCA2, and HOXA9. Unfortunately, BRCA2 and HOXA9 peptides were not identified. We did find RAD51 expression was elevated in diagnostic and relapsed B-ALL samples (Reviewer Figure 1B), which is consistent with elevated DNA damage secondary to HR deficiency, as seen in the AML models that were sensitive to PARP1 inhibition (Esposito et al 2015). However, we were not able to correlate RAD51 expression with IC50 values for PARP inhibitors as RAD51 was not universally identified in all samples that we tested for sensitivity to PARPi.

Major comments:

1) Authors claim that the set of targetable lesions persists in pediatric ALL samples they profiled. However, in their validation set, e.g. Figure 4G, no matched diagnosis relapsed samples were tested for their sensitivity to Olaparib or PJ34. Formally, it is possible that sensitivity to this therapy evolves in the matched patients despite similar protein expression level.

Response. The reviewer is correct that our original assessment of sensitivity to PARP inhibitors did not include matched progression samples. We re-engaged our Biobank to identify viably-frozen matched progression samples with sufficient leukemia cell content for additional drug response analysis. We identified matched progression samples from two patients, also with next-generation sequencing data and proteomes.

We sourced additional viably frozen samples for the Dx timepoint that match the R timepoints already included in the manuscript (BALL04 Dx and BALL06 Dx) and an additional two samples that were included in the previous experiment as a control (BALL04 R1 and BALL03 R2). Upon thawing and attempting the PARP inhibitor experiments on BALL06 Dx and BALL03 R2, the viability was exceptionally low (mean viability in DMSO at 33.9% and 29.8% respectively) and below our established cut-off of viability >45%.

Although an analysis of matched progression samples would be ideal, and is a potential future line of investigation into the possible evolution of PARP inhibitor therapy response, we do not have access to a sufficient number of the needed samples: matched Dx-R progression samples that are viable cryopreserved. However, we are confident in our proteome-based discovery of PARP1 as a molecular target in Dx and R B-ALL. Importantly, PARP inhibitors – agents against this proteome-identified pan ALL target - show more selectivity than the other molecular-targeted agents we tested, which were identified based on gene mutation or copy number variation.

2) More detailed discussion of the additional possible targets are welcomed.

Response. We thank the reviewer for this important suggestion. In our revised manuscript, we expand the discussion to more fully detail additional possible targets identified via proteomic analysis. The highest ranked targets that we identified (Figure 4C) include HSPB1 (aka HSP27), ANAPC1, and PRDX1. We focus our discussion on targeted inhibition of HSP27 and PRDX1.

Minor comments:

1) Lane 454, what is GFP viability dye?

Response. In the revised manuscript, we clarify our methods and replace “GFP viability dye” with CyQUANT Direct (Green) (Thermo Fisher Scientific).

2) Figure 1G, the coloring does not fit the legend.

Response. We removed the green color in Figure 1G.

Reviewer #3, expertise in DNA damage, pediatric ALL and drug development (Remarks to the Author):

Lorentzian et al present a comprehensive retrospective analysis of paired ALL diagnosis and relapsed specimen. They argue that a major challenge for prospective precision oncology approaches is a limited understanding of the persistence or evolution of targetable lesions and their associated proteins or pathways, and responses to targeted agents, that may be gained or lost at relapse. Consequently, they suggest that because proteins are the actual therapeutic targets for most oncology drugs, it is imperative to understand the pediatric tumor proteome to determine how the response to therapy may change through progression.

1. I applaud the authors for their effort to interrogate the proteome in diagnostic and relapsed ALL samples, and I found some of their findings intriguing. However, I have a fundamental disagreement with what I believe is a central premise in this work.

Response. We are grateful for the reviewer’s comments and insights. However, we feel that our data, and central premise, agree (rather than disagree) with the reviewer’s comments, as we detailed in the following.

2. From my experience in analyzing both RNA and protein expression experiment, the number of differentially expressed genes or proteins induced by therapeutic intervention is relatively small compared to the number of genes that vary due to cell identity. In PCA analysis, the first principal component often discriminates between different cell models – meaning, variation due to treatment is relatively small in magnitude. So, how much should we expect the proteome to change from diagnosis to relapse? Most of the proteins—including cancer associated proteins—are probably not consequential to a specific therapeutic response. In fact, for targeted therapy, a single mutation can disrupt drug binding and cause resistance to therapy, and this change could be difficult to detect from proteomic data alone. In the Ma et al paper cited by the authors, while there was considerable overlap in variants between diagnosis and relapse, six pathways were frequently mutated, with NT5C2, CREBBP, WHSC1, TP53, USH2A, NRAS and IKZF1 mutations enriched at relapse. In fact, relapse-specific mutations in NT5C2 were relatively common, and it is known that mutations in this gene confer resistance to certain ALL treatments. Therefore, the distinction between diagnosis and relapse could be subtle. It would be helpful for the authors to acknowledge these findings from the Ma et al paper.

Response. We thank the reviewer for these important points. It is well-established that childhood leukemia clones evolve from diagnosis through treatment to relapse (Li et al, Blood 2020, Andersson et al, Cancer Research 2020, Hunger & Mullighan 2015). This evolution predicts the clone(s) that evolve the relapse will differ from the diagnostic clone(s). Thus, this dogma predicts the initiation of precision oncology approaches at diagnosis – a time when standard of care therapies are most frequently applied (as

opposed to more targeted approaches) – would most likely not be informative for relapsed disease. However, our data indicates a stability in targetable lesions that supports the application of precision approaches initiated at diagnosis.

As the reviewer mentions, however, the distinction between diagnosis and relapse could be subtler with clones distinguished only by therapy-driven mutations (such as frequently observed *NT5C2* mutations). Indeed, our data, and our interpretation of that data, is in complete agreement with the reviewer's assessment, and in agreement with the findings outlined in the Ma et al paper. Thus, in our revised manuscript, we include the Ma et al paper in the introduction as it was in our original manuscript. In addition, and as recommended by reviewer 3, we expand our discussion of their dataset (lines 117-119, 341-345) to more clearly address the reviewer's points.

These new sections read as follows:

“One challenge to initiating precision medicine at diagnosis is the prospect that the dominant relapse clones contain distinct mutations and unique drug sensitivities. Genomic analysis of paired B-ALL samples has inferred clonal structure and evolution through disease progression (3,27–29); thus, for example, a transition from a major clone at diagnosis carrying a *KRAS.G12D* mutation is distinguishable from a major clone at relapse carrying a *KRAS.A146T* mutation (3). Our analysis did not focus on the gain/loss of clone-defining mutations. Rather, we measured the durability of actionable genomic mutations and stable proteomic features over the course of the disease; in the example of a transition for *KRAS.G12D* to *KRAS.A146T* highlighted above, for instance, our analysis did not distinguish between site-specific mutations in the same actionable target.”

3. Furthermore, are there data sets comparing gene expression between diagnostic and relapsed ALL tumors? Are the number of genes that change small as in this study?

Response. We performed NEW whole transcriptome analysis on 4 matched pediatric ALL progression pairs sourced from the BC Children's Hospital biobank. Three of the four pairs clustered together based on Pearson correlation. The one pair that did not also had the lowest genome and proteome stability. To increase the power of this study, we analyzed a whole transcriptome data set of 35 matched pediatric ALL progression pairs (70 samples) sourced from the TARGET initiative (Vaske et al, JAMA 2019). In this new data, hierarchical clustering based on Pearson correlation of all samples showed 34% of the pairs clustered as neighbors, while an additional 17% of pairs were within the same major cluster (new Supplementary Figure S11B).

For each pair Dx vs R, Pearson correlation of cancer associated gene products also demonstrated high stability with 83% of pairs having an r value $>.85$ on the transcript level (new Supplementary Figure S11C), consistent with our protein level analysis (Figure 3F). In the BCCH cohort, two pairs that demonstrate very high stability at the genome and global proteome and CAP proteome level also demonstrated highest stability at the transcript level (BALL01 $r=0.94$, BALL04, $r=0.98$), while the patient that had the least stability overall had the lowest CAP stability (BALL06 $r=0.7$). These findings confirm that RNA has a similar pattern of stability between Dx and R (or R-R) that we observe with the genome and proteome.

4. The authors then go on to identify PARP1 as a protein that is over-expressed in B-ALL compared to non-cancer bone marrow-derived stem cells, and whose expression remains consistently elevated in diagnosis and relapse. However, PARP1 plays a role in the cell cycle and its expression might be higher in cycling cells (see PMID 24367566 and 29886395). Thus, is the increase in PARP1 expression in B-ALL simply a reflection of a higher proliferation rate?

Perhaps this could explain why in Figure 4F, B-ALL cells had higher levels of nuclear PARP1 compared to BMSC independent of genotoxic stress at the 24 h timepoint (there seems to be a bimodal distribution as well – perhaps cell cycle related?). Likewise, while PARP inhibitors appeared to be more effective in B-ALL cells compared to BMSC cells, could this simply be a function of higher growth rate, as cycling cells are more prone to DNA damage and PARP1 is important for the DNA damage response? One possible way to address the effect of proliferation would be to repeat the experiment in Figure 4H with Topo1 inhibitors, which are also selective for proliferating cells, and anti-mitotic agents like vincristine. Finding a large therapeutic window for PARP inhibitors, but not the other agents, would strongly support the author's rationale to target PARP1 in B-ALL.

Response. In our revision, we investigate the excellent key point that PARP sensitivity might be an effect of increased proliferation. In our new data, we demonstrate that less than 2.5% of primary cells are proliferating within the in vitro drug screening assays, as measured by phospho-histone H3 immunofluorescence (Supplementary Figure S13A,B).

To more directly address the reviewer's key point, we assessed the expression of 14 cell cycle proteins previously found to peak in abundance in G2 and M relative to G1 phases via proteomics of intracellular immunolabelled cell subsets analysis (Ly et al, eLife 2017). While PARP1 expression was uniformly elevated in diagnostic and relapse samples, relative to non-cancer bone marrow samples, none of the 14 cell cycle regulated proteins showed a similar uniform elevation, and none of these proteins correlated to PARP1 levels (Supplementary Figure S13C). Thus, we are confident that elevated PARP1 expression in diagnosis and relapse samples is not simply a surrogate read-out for samples with an elevated rate of proliferation, as we do not observe elevated G2/M phase proteins in these samples. This new data is described in lines 309-313.

These new sections read as follows:

“To investigate whether this result is an effect of increased cell proliferation, we measured the percentage of phospho-histone H3 positive cells in culture. We found, however, <2.5% of cells in the in vitro drug screening assays are mitotic (Supplementary Fig. S13A-B) suggesting cell proliferation is not the target for PARP inhibition. We next measured the protein abundance of key mitotic and cell cycle regulators shown to differentiate cells in G2 and M phases (1). While the expression level of PARP1 was universally higher, we found the levels of key mitotic and cell cycle regulators were not elevated in Dx and R samples relative to non-cancer BM controls (Supplementary Fig. S13C,D).”

While I agree the proteome data set is interesting and worth reporting, these points above dampen my enthusiasm for the paper. Moreover, it would be helpful to clarify the following

points for the reader:

5. In Figure 1B, it is difficult to interpret the Circos plots. Perhaps it would be better to represent this data in a heatmap with variants depicted as circles whose radius indicates frequency of occurrence and with color representing degree the variant is shared in diagnosis and relapse?

Response. We appreciate the reviewer’s comment and we plot the data below as suggested by the reviewer (Reviewer Figure 2). But, we feel a Circos plot more clearly illustrates the retention of variants, and we have left those Circos plots in the manuscript.

We have additional plots that also represent the genomic lesions and lesion stability Supplementary Figures 2 and 3.

Summary of mutations detected in the BCCH cohort. The size of the circle represents the number of times the mutation occurred in the data set and the color of the circles represents the fraction of occurrences in which the mutation was shared between Dx and R

6. For Figures 1B-1E, what is the set of variants analyzed? Is this the set of all somatic mutations identified across all samples? Or somatic mutations reported as cancer drivers?

Response This set is all somatic mutations identified across all samples. We added lines 98-99 and 108-109 to more clearly state this.

These new sections read as follows:

“We first explored the mutational landscape in paired progression samples (n=10 B-ALL, n=1 T-ALL) in the BCCH cohort via targeted, pediatric cancer-focused NGS analysis and reported all somatic mutations identified”

“To determine the generalizability of this finding, we mined all mutational findings from an additional cohort.”

7. In Figure 1C, wouldn't one matched pair be represented as a point in each of the three groups (Dx, R, and Share), with the 3 points summing to 1.0? The data is plotted in a way that loses the relationship between matched sample pairs. Please consider presenting the data as a stacked barplot (as is done in the supplement): this would more clearly show the variation in shared vs unique variants found per sample.

Response The data has been plotted as suggested and added to Supplementary Figure S2.

8. In Figure 1D-E, the relationship between % shared and tumor subtype is striking. It would be helpful to confirm that these differences are statistically significant using the appropriate tests (maybe ANOVA?).

Response We thank the reviewer for the suggestion. We performed an ANOVA to test for significance but none of the comparisons were significant, likely due to high variance and low n in many groups. In the Figure 1 legend, we state that the test was performed but the result was not statistically significant.

9. Figure 1F is a nice way of showing variation between diagnosis and relapse. It is striking that for each target, there are examples of variation between diagnosis and relapse. This is most evident for the CDK4/6, MEK, and MTOR pathways. Is the nature of variants different in those that are consistent vs. those that are variable?

Response We agree that this is an interesting finding. We have added lines 120-121 to address this point.

These new sections read as follows:

“The targets with the highest retention were CDKN2A deletions paired with CDK4/6 inhibitors (shared in 68.9% of occurrences), and NRAS and KRAS mutations paired with MEK inhibitors (shared in 57.1% and 61.5% of occurrences respectively), although NRAS mutations occurred in 21 patients compared to 13 for KRAS (Fig. 1F).”

10. In Figure 1G, I would suggest representing EC50 on a log scale, especially for the MEK plot where EC50 values range at least two orders of magnitude. It likely that this would make the difference between target positive and negative groups in the MEK plot look much smaller. The plot in H, which is on the log scale, clearly indicates close agreement between diagnosis and relapse.

Response We have changed the axis to be in a log scale.

11. In Figure 2C, the authors conclude that proteome analysis shows stability through progression. However, there were only 6 matched samples in this analysis, and 2 of the 6 clustered diagnosis and relapse in different groups. This conclusion is not supported by the data.

Response The trend identified in Figure 2C is interesting but, as the reviewer points out, does not support a conclusion that proteomes are stable. The observed trend, however, seeded our subsequent analyses, in which we use an Equivalence test (Figure 3A) and we then analyse the stability of cancer associated proteins (CAPs) (Figure 3E,F).

The paired Dx-R samples had the highest percentage of statistically equivalent proteins between pairs and this finding was significant compared to other random pairings (mann-whitney U test p-value <0.001) (Figure 3A). The direct comparison of CAPs in paired Dx-R samples demonstrated high correlation of abundance, with five of the seven pairs having a Pearson's r correlation value >0.75. Together, these data support the conclusion that the paired samples have stable proteomes.

The two pairs mentioned by the reviewer (BALL05 and BALL06) also have the highest divergence for genetic lesions (Figure 1F), whole proteome stability (Figure 3A), and CAP stability (Figure 3F). Thus, it is consistent that these two samples cluster in different groups.

12. The survival analysis in Figures 2D-E is interesting. There were no events in P2, P3, and P7 – is this correct? P4 was the cluster showing the poorest outcome, but this cluster was also enriched for BCP-ALL and HR tumors. Would the analysis of the clusters P2, P3, P4, and P7 based on risk group (not cluster group) result in a significant difference in outcome? Is the result largely driven by BCL-ALL?

Response Cluster P2, P3, and P7 had 1, 1, and 2 events, respectively (Figure 2C). Cluster P4 is enriched for BCP-ALL and HR tumors, but one patient in cluster P4 was SR and relapsed later than 5 years. Provocatively, this patient is the only one in the P4 cluster that did not have an event in the 5-year EFS (as the relapse occurred after 5 years).

A five-year survival analysis of clusters P2, P3, P4, and P7 based on risk group does not yield a significant difference in outcome (Reviewer Figure 3).

5 year survival by risk group: Clusters P2,3,4,7

Kaplan Meier survival curve with up to 5 year follow-up data for all samples grouped clinically assigned risk group (SR= standard risk, HR= high risk) for only clusters P2,P3, P4, and P7 , ns= not significant by unpaired t-test. Black tick marks on the survival curve represents data that has been censored due to follow-up data <5 years.

13. It's difficult for me to follow the statistical analysis performed in Figure 3B. Did the authors look at all significant differentially expressed proteins between matched diagnosis and relapse,

and identify those which are consistently different (meaning enriched in diagnosis or enriched in relapse across all samples)? Only equivalent proteome and relapsed-enriched proteome is presented. Shouldn't there also be a diagnosis-enriched proteome? Would GSEA on the non-equivalent proteome, restricted to proteins that change with large logFC in the same direction for diagnosis and relapse, be informative?

Response The reviewer brings up an interesting point to better interrogate the differences between Dx and R. To this point, we performed a new analysis (Supplementary Figure S#11) to directly compare the proteins that are different between Dx and R for each patient. The new heatmap shows the log₂FC (Dx/R) for all of the proteins that were statistically significantly different (student's t-test FDR <0.05, log₂FC >1 or log₂FC <-1), in all pairings. Indeed, the differences appear to be patient specific. We attempted an analysis of a Dx-enriched proteome and a R-enriched proteome based on proteins that had a high log₂FC (>1 or <-1) in at least 3 pairings to determine more general relapse enriched processes, however in this analysis no biological process showed significant enrichment for either the Dx or R. Note that the statistical analysis in the previous version of the manuscript has been altered to correct for FDR and results for this analysis are now slightly different. Lines (235-239) have been added to better explain this analysis in the manuscript.

These new sections read as follows:

“Investigation of proteins that were statistically different between the pairs did not reveal any significantly enriched processes. Alternatively, a heatmap of the log₂FC of timepoint 1 (T1)/ timepoint 2 (T2) of all of the statistically different proteins demonstrates that the differences between Dx-R are patient-specific (Supplementary Fig. S11, Supplementary Table S14)”

14. In Figure 3F, while it is true that there is a significant correlation between diagnosis and proteome, in six of the seven cases, there are clear outliers. What are these? Perhaps they have biological significance?

Response This indeed is an interesting point. Many of the clear outliers are those that were detected in one timepoint but not detected in the other timepoint (and imputed). FLT3 was consistently high in many of the Dx (or T1) samples but low or not detected in the Relapse (BALL01, BALL02, BALL03, BALL05) and the opposite trend in BALL07. Other proteins were patient specific such as SELP in BALL04. Lines 261-265 have been added to address this.

These new sections read as follows:

Further investigation of the few outliers revealed that FLT3 was commonly over-abundant at Dx and lower abundant at Relapse (observed in BALL 01R4-R5P, BALL02, BALL03, BALL05) suggesting the loss of FLT3 may be a relapse-specific mechanism. Other outliers were patient-specific, such as the higher abundance of SELP at Dx for BALL04.

Reviewer #4, expertise in proteomics (Remarks to the Author):

In the manuscript titled “Persistence of targetable lesions, predicted therapy sensitivity and proteomes through disease evolution in pediatric acute lymphoblastic leukemia” authors applied genomics and proteomics to examine, proteogenomic evolution in relapsed pediatric acute lymphoblastic leukemia patients. In my reviewing, the manuscript in its current form is not suitable for publication in Nature communication.

Major concern:

1. The manuscript is not coherent, at times it is difficult to follow the correlation between different sections of results. The manuscript lacks a clear finding and novelty. The rationale for using different proteomics analysis presented in the manuscript is not clearly defined. It is also not clear how genomics and proteomics are compared, whether or not same patient cohort was used for proteomics and genomics analysis. Overall manuscript lacks clarity, flow of the story and novel findings.

Response We appreciate the reviewer's time to evaluate our manuscript and identify aspects that can be further strengthened. The reviewer highlights several issues that we have now resolved.

As noted in our responses to reviewer 1, reviewer 2 and reviewer 3, we have added a new Supplementary Figure 1 to graphically depict the analyses performed on each sample that we sourced from the BCCH biobank. We have also revised the manuscript to clarify our findings and to clearly outline the rationale for the proteomics analysis that we performed.

Reviewer 4 states that our manuscript lacks a clear finding and novelty. We strongly disagree with this statement. As well, the comments of reviewer 1, reviewer 2, and reviewer 3 (below, underlines added) specify our main novel finding of durability of actionable mutations and proteomes, and each reviewer highlights the novelty and importance of our research. Nevertheless, we have edited our manuscript to more clearly state the novelty and improve the flow of the story, in response to the comments of Reviewer 4.

R1 - “their main finding of high rates of durability of actionable genomic mutations and stable proteomic features over the course of the disease would seem to have important medical implications. This is an extremely timely topic of interest for readers and certainly within the journal's scope”

R2- “This study supports the notion that proteomes of the malignant cells in B-ALL are stable during the disease and provides a proteomic analysis-based pipeline for the potential therapeutic targets discovery. Importantly, authors also experimentally demonstrated poor selectivity towards targeted therapies (e.g CDK4i, MEKi and more) based on the genetic analysis alone. In my view, this work delivers significant technological and bioinformatics accomplishments for the field of personalized medicine and pediatric oncology. The work fits well with the established criteria in the literature of precision medicine based on proteogenomic approaches.”

R3- “They argue that a major challenge for prospective precision oncology approaches is a limited understanding of the persistence or evolution of targetable lesions and their associated proteins or pathways, and responses to targeted agents, that may be gained or lost at relapse. Consequently, they suggest that because proteins are the actual therapeutic targets for most oncology drugs, it is imperative to understand the pediatric tumor proteome to determine how the response to therapy may change through progression. I applaud the authors for their effort to interrogate the proteome in diagnostic and relapsed ALL samples, and I found some of their findings intriguing.”

2. Authors used different cohorts for different analysis, and it is not clear how each cohort is related to each other and how they were compared between each other.

For example, in section “Next-generation sequencing (NGS) reveals stability of affected genes through ALL disease progression” authors writes , “we sourced 71 bone marrow biopsies (n=44 ALL at initial diagnosis (Dx), n=21 ALL at relapse (R), n=6 non-cancer bone marrow) from pediatric patients seen at BC Children’s Hospital (BCCH) (Supplementary Table S1) and publicly available whole-exome sequencing (WES) data from 138 specimens collected and analyzed by the St. Jude’s Children’s Research Hospital (SJH)” it is not clear how the 71 bone marrow biopsies data and 138 whole-exome sequencing data is used/compared?

Response We agree with the reviewer’s assessment, and we have re-written this paragraph to more clearly state the samples that are specific to the analysis and results being presented.

These new sections read as follows:

“To examine genomic evolution in relapsed pediatric ALL, we sourced 25 paired initial diagnosis (Dx) and relapse (R) bone marrow biopsies from 11 pediatric patients seen at BC Children’s Hospital (BCCH) (Supplementary Table S1) and publicly available whole-exome sequencing (WES) data from 138 specimens (69 paired biopsies) collected and analyzed by the St. Jude’s Children’s Research Hospital (SJH) (Fig. 1A, Supplementary Fig. 1).”

3. Again, in line 97 author writes “We first explored the mutational landscape in paired progression samples (n=10 B-ALL, n=1 T ALL) via targeted, pediatric cancer-focused NGS analysis (10).” It is not clear whether samples in this cohort are part of which cohort or is it a separate cohort?

Response Again, we agree with the reviewer’s assessment. To address this comment, we have re-written this sentence and we have added a new Supplementary Figure 1 to clarify the analyses performed on each sample from the BC Children’s Hospital (BCCH) cohort.

4. Again, in line 107 author writes, “To determine the generalizability of this finding, we mined public NGS datasets from ALL cases (n=49 B-ALL; n=20 T-ALL) treated at St. Jude’s Hospital (SJH).” It is not clear whether samples in this cohort are part of which cohort or is it a separate cohort?

Response We have clarified that the St. Jude’s Hospital cohort is a separate cohort from the BC Children’s Hospital cohort, and that we examined the St. Jude’s Hospital (SJH) cohort to determine whether our findings of stability in the BCCH cohort are generalizable to another distinct cohort.

5. Again, in line 125 author writes “In the SJH and BCCH cohorts (n=80 paired samples)” it is not clear which 80 patients author is referring to?

Response We have clarified that the 80 patients are a combination of the BCCH cohort (n= 11 patients) and the SJH cohort (n= 69 patients).

6. Again, in section “Global proteome analysis shows stability through progression and groups cases with poor outcome” in line 153 author writes “we next conducted a comprehensive analysis of 48 primary (n=39 B-ALL, n=9 T-ALL) specimens from Dx and R, including 14 specimens from 6 patients with matched biopsies taken at diagnostic and subsequent relapse timepoints. It is not clear whether samples in this cohort are part of which cohort or is it a separate cohort?”

Response We have re-written this sentence for clarity and we have added a new Supplementary Figure 1 to clarify the analyses performed on each sample from the BC Children’s Hospital (BCCH) cohort.

7. In line 165, author writes “Samples clustered distinctly by B-ALL, T-ALL and non-cancer monocytes and cell lines (Fig. 3B).” However, Fig.3B in the manuscript represents pathway enrichment analysis and not cluster analysis. Please correct it.

Response We thank the reviewer for pointing out this typo in the manuscript. This sentence is in fact referring to Figure 2B which is a clustering analysis. This typo has been corrected in the revised manuscript.

8. Also, I would suggest adding a PCA plot based on all quantified proteins in each group to show the separation of different groups of samples. I would also suggest author to provide a PCA cluster analysis based on age and sex of the patients to show the effect of age and sex if any.

Response We appreciate the suggestion for this additional informative analysis of our data. We conducted PCA analysis to demonstrate the separation of samples based on sample type and the results are presented in NEW supplementary Figure S7. Based on PC1 and PC2 the samples separate by ALL, non-cancer BM and cell line (Supplementary Figure S7A). PC3 and PC4 further separate the ALL group into B-ALL and T-ALL (Supplementary Figure 7B). The symbols represent sex to demonstrate there are no obvious biases based on sex. Additionally, we investigated the contribution of each metavariable per PC using the Kendall correlation method and produced a correlation heatmap (Supplementary Figure S7C). We conclude that age and sex do not contribute significantly to the variation in the data and the variation is most driven by sample type (B-ALL, T-ALL, Non-Cancer BM, or Cell Line), addressed in lines 173-175.

9. In Figure 2B and 2C, it is not clear what the fold change refers to, meaning how the fold change is calculated between different sample type/cytogenetic group? Also, for different groups there are different numbers of samples, and different n could have an impact on the statistical analysis. I would like authors to comment if they evaluated the effect of size on fold -change significance.

Response We now describe how the log₂FC was calculated (lines 167-169). Because the fold change for each sample is calculated by the protein abundance/median protein abundance across all samples, different group sizes have no effect. In addition, we performed no calculations within specific groups and assigned no significance based on log₂FC values.

10. In Fig.4A it is not clear why a random cutoff for log₁₀ intensity at 6 is selected. There are

many other proteins which have higher fold change and higher intensity than PARP1 and those proteins are not discussed.

Response We appreciate the reviewer's comment, and have now clarified that the cut-offs were assigned based on the top 95th percentile of highest abundant proteins, which was a log₁₀ intensity of 6. Conversely, the 95th percentile for FC over normal was a log₂FC of 1.7.

The reviewer correctly points out that other proteins were identified with a higher fold change and higher intensity than PARP1. So, we expanded the revised discussion to more fully detail these additional possible targets, which include HSPB1 (aka HSP27), ANAPC1, and PRDX1 as the highest ranked targets (Figure 4C). We discuss targeted inhibition of HSP27 or PRDX1 as a potential future direction of research.

Reviewers' Comments:

Reviewer #1:

Remarks to the Author:

The authors have addressed my concerns.

Reviewer #2:

None

Reviewer #3:

Remarks to the Author:

The revised manuscript is substantially clearer and better supported. My concerns have been sufficiently addressed.

Reviewer #5:

Remarks to the Author:

The authors have addressed all comments from reviewer #4 adequately.

REVIEWERS' COMMENTS

Reviewer #1 (Remarks to the Author):

The authors have addressed my concerns.

Response: We are thankful for the reviewer's careful review of the manuscript and for the suggestions that have improved the manuscript.

Reviewer #3 (Remarks to the Author):

The revised manuscript is substantially clearer and better supported. My concerns have been sufficiently addressed.

Response: We are thankful for the reviewer's careful review of the manuscript and for the suggestions that have improved the manuscript.

Reviewer #5, with expertise in technical proteomics to replace Reviewer #4 (Remarks to the Author):

The authors have addressed all comments from reviewer #4 adequately.

Response: We are thankful for the reviewer's careful review of the manuscript and for the suggestions that have improved the manuscript.